# A³-GS: ANIMATE ANY ARTICULATED OBJECTS WITH 3D GAUSSIAN SPLATTING

## ABSTRACT

3D Gaussian representations have demonstrated remarkable success in reconstructing complex scenes and rendering novel views. However, rigging and animating template-free articulated objects represented by 3D Gaussians remain challenging. A key difficulty arises from the discretization of Gaussians, which often produces artifacts during animation. Another challenge lies in rigging arbitrary, template-free shapes without sufficient training data. To address these challenges, we propose *A³-GS*, a new 3D Gaussian splatting-based framework that reconstructs articulated objects from multi-view images while simultaneously estimating a skeleton and skinning weights, thereby enabling rich animations with high-quality rendering. Technically, we first introduce a Mesh–Gaussian hybrid representation for articulated objects. By exploiting the continuity of the mesh, our method mitigates rendering artifacts such as spikes and tearing caused by Gaussian deformation, thereby enhancing the visual quality of animated results. We further learn motion-coherent skinning weights by leveraging motion priors from visual foundation models trained on large-scale 2D video data, reducing reliance on scarce 3D datasets. In addition, we incorporate local rigidity regularization to improve the smoothness of skeleton-based deformation and further suppress artifacts. Extensive experiments validate the effectiveness of our approach, demonstrating clear advantages over existing methods. The code will be made publicly available.

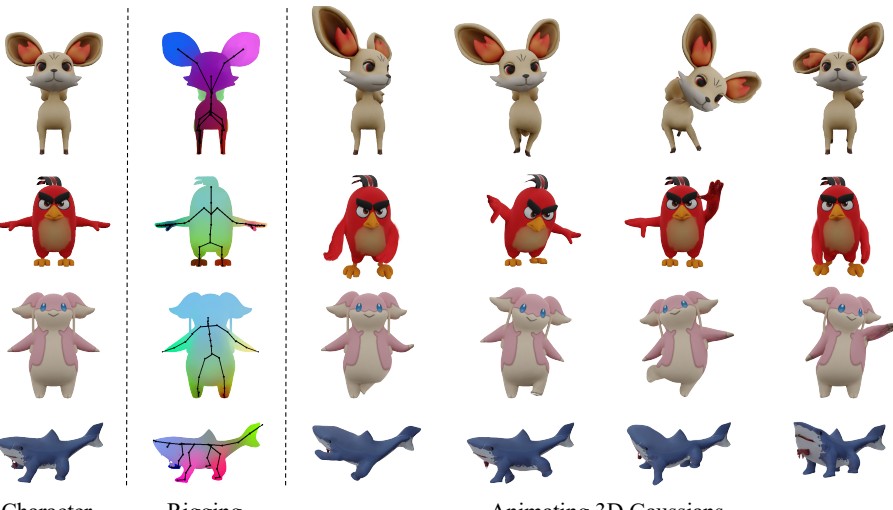

Figure 1: A³-GS reconstructs articulated objects from multi-view images with a Mesh-Gaussian hybrid representation and a learned rig, resulting in realistic animation with high-quality rendering.

## 1 INTRODUCTION

3D content generation is a key challenge in the fields of computer graphics and 3D vision, enabling a wide range of applications in gaming, filmmaking, VR/AR, and robotics. Reconstructing 3D ob-

jects, generating skeletons, along with binding and animating them with high-quality free-viewpoint rendering, forms a fundamental aspect of 3D dynamic content generation.

Articulated objects are typically represented using meshes. While meshes are highly flexible, they suffer from limited rendering quality. Recently, 3D Gaussian has gained popularity, offering high-quality rendering and real-time performance. However, its overly discretized nature makes it prone to artifacts during animation. These defects are mainly caused by the approximation error of the deformed Gaussians. Some methods introduce additional information, such as meshes or local tetrahedra (Gao et al., 2024; Ma et al., 2024; Jiang et al., 2024), to solve the problems in Gaussian editing. Inspired by them, we introduce mesh into the 3D Gaussian representation to improve the rendering quality during animation.

Traditional rigging pipelines in the industry often rely on complex and time-consuming manual operations. Automating the process of rigging for articulated objects remains a challenging task. Compared to sparsely structured skeletons, manually painting skinning weights will be more time-consuming, so automated skeleton prediction is a more critical task. Early methods (Baran & Popovic, 2007; Kavan & Sorkine, 2012; Dionne & de Lasa, 2014) rely on local surface properties and the spatial distance between the skeleton and the surface to optimize skinning weights. Since they rely solely on geometric attributes without incorporating semantic information, the performance is limited. In recent years, numerous data-driven methods have emerged that leverage deep learning techniques to predict skinning weights with ground-truth supervision (Liu et al., 2019; Xu et al., 2020; Mosella-Montoro & Ruiz-Hidalgo, 2022; Zhang et al., 2025b; Song et al., 2025b). However, due to the difficulty of obtaining high-quality ground-truth data, the available training datasets are often limited in quantity and vary greatly in quality. Some methods (Xu et al., 2022; Liao et al., 2024; Uzolas et al., 2024; Wan et al., 2024; Yao et al., 2025) utilize motion priors to estimate skinning weights, but they are usually limited by the motion provided. The core problem lies in the insufficient amount of 3D data with ground-truth rigging or dynamic motion sequences. Therefore, how to automatically estimate high-fidelity skinning weights for static objects in the absence of sufficient training data remains an open research challenge.

To address these challenges, we propose $A^3$-GS, a framework that reconstructs 3D static objects from multi-view images, extracts skeletons, and predicts skinning weights, thereby enabling realistic animation with high-quality rendering, as shown in Fig. 1. To ensure rendering fidelity during animation, we adopt a Mesh-Gaussian hybrid representation: it leverages the efficiency and realism of 3D Gaussians, while exploiting mesh continuity to mitigate artifacts arising from Gaussian deformation. Recognizing the abundance of 2D video data and the strong capabilities of 2D generative models, we incorporate motion priors from these models to enhance rigging learning under limited 3D data. Specifically, we adopt a drag-based image editing model (Shi et al., 2024) as the base model. By driving skeletons into novel poses, joint movements serve as control signals, along with the object's static image, to generate new posed images. These synthesized images then provide guidance for learning skinning weights. Since some generated images may suffer from poor quality or physical implausibility (e.g., unnatural limb counts), we incorporate the drag signal design strategy and training techniques. In addition, we introduce the mesh-based as-rigid-as-possible (ARAP) regularization term to improve the robustness of training and local smoothness of skin deformation. Extensive experiments validate the effectiveness of the proposed method.

In summary, our contributions are as follows.

- We propose a new framework with a Mesh-Gaussian hybrid representation to reconstruct articulated objects from multi-view images with learned rig, enabling realistic animation with high-quality rendering;

- We are the first to propose using a drag-based image editing model to learn skinning weights, using the changes in joint projections to guide the generation of corresponding images, and thus guide the learning process of skinning weights;

- We design effective training strategies to mitigate the impact of generating unrealistic images and incorporate an ARAP regularization term to enhance the robustness of skinning weight learning.

## 2 RELATED WORK

**3D Representation for Animated Objects.** To model animatable objects, traditional pipelines typically use meshes to represent the object and establish a skeleton and skin weights for it. While

mesh-based representations are flexible and mature, their rendering quality is limited. Some approaches incorporate the higher-quality rendering model NeRF (Mildenhall et al., 2020) for animatable object modeling (Uzolas et al., 2023; Zhao et al., 2022). However, such NeRF-based techniques often suffer from high computational costs in both training and rendering, especially for high-quality objects. Recently, 3D Gaussian Splatting (Kerbl et al., 2023b) has gained significant attention for achieving comparable visual quality while enabling real-time rendering performance. However, owing to its highly discrete nature, the reconstructed models tend to exhibit artifacts when being edited or animated. To improve the performance of Gaussian scene editing, GaussianMesh (Gao et al., 2024) binds the 3D Gaussians to the mesh facets, exploiting the continuity of the mesh to model the deformed Gaussian. VR-GS (Jiang et al., 2024) embeds each Gaussian into a tetrahedron, approximating the deformation as piecewise linear. ARAP-GS (Han et al., 2025) adopts k-nearest neighbors and local rigidity constraints to estimate the deformed Gaussians. For articulated objects, some methods (Wan et al., 2024; Moreau et al., 2024) estimate the covariance of deformed Gaussians through Linear Blend Skinning (LBS)-based computation. RigGS (Yao et al., 2025) notes that significant deformation can locally introduce artifacts, thus adopting isotropic Gaussians to reduce these imperfections. However, this also compromises rendering quality.

**Skeleton Generation.** Generating semantically skeletons for static objects is a challenging task. It requires accurate positioning of joint points and establishing reasonable topological relationships. Some methods utilize shape priors to manually design a general skeleton for a class of objects, such as SMPL (Loper et al., 2015) for human bodies and SMAL (Zuffi et al., 2017) for quadrupeds. Since a wide range of shapes cannot be uniformly defined in advance, data-driven methods have been proposed to solve this problem. Xu et al. (2019) converts the input 3D shape into a voxel representation and predicts the skeleton by combining geometric shape features. RigNet (Xu et al., 2020) predicts the positions of joints based on a graph neural network and attention-driven clustering, and proposes BoneNet to predict the probability of links between joint points to obtain a complete skeleton. DRiVE (Sun et al., 2025a) proposes 3D Gaussian-based diffusion network, to accurately predict joint positions. Anymate (Deng et al., 2025) contributes a big rigging dataset including multi-category objects, and tests regression-based, diffusion-based, and volume-based architectures for joint prediction. Recently, with the popularity of the autoregressive model, some methods (Liu et al., 2025; Zhang et al., 2025b; Sun et al., 2025b; Song et al., 2025b;a) adopt it to predict skeletons. However, the insufficient quality and quantity of datasets limit their performance.

**Skinning Weight Learning.** Unlike sparse, intuitive, and potentially predefined skeletons, automated prediction of skinning weights would be more practical. Baran & Popovic (2007) use of Laplace's diffusion equation to generate weights. Dionne & de Lasa (2013) voxelizes the input shape and estimates the skinning weights by computing volumetric geodesic distances. NeuroSkinning (Liu et al., 2019) constructs a graph for the input mesh and skeleton, assigns mesh-skeleton attributes to each graph node, and feeds them into the graph convolution network to predict skinning weight. SkinningNet (Mosella-Montoro & Ruiz-Hidalgo, 2022) introduces a two-stream graph convolutional architecture to extract features from the input mesh and skeleton, and adopts multi-aggregator Graph Convolution (MAGC) layer to predict skinning weights. Make-It-Animatable (Guo et al., 2025) proposes a particle-based autoencoder and structure-aware modeling strategy to improve accuracy and robustness. HumanRig (Chu et al., 2025) presents a mesh-skeleton mutual attention network to predict a refined skeleton and skinning weights with a joint learning strategy. MagicArticulate (Song et al., 2025b) predicts skinning weights based on the functional diffusion framework. UniRig (Zhang et al., 2025b) leverages a bone-informed cross-attention mechanism to predict skin weights and bone attributes. Puppeteer (Song et al., 2025a) incorporates part-aware features and presents an attention-based network to predict per-vertex skinning weights. Similar to skeleton generation, the biggest bottleneck of these methods lies in the quantity and quality of training data. The high cost of acquiring 3D data, especially high-quality 3D motion data or data with ground truth, limits the generalization performance of these methods.

**Generative Priors for Animated Objects.** Compared with the scarcity of 3D dynamic data, 2D videos are massive, which has promoted the rapid development of 2D video generative models. Some methods introduce visual foundation models for animated objects. For rigged 3D models, Li et al. (2025) employs 3D Gaussian as the fundamental representation and utilizes SDS loss to train text-driven 3D motion generation based on the CogVideoX-5B model (Yang et al., 2024). Puppeteer (Song et al., 2025a) leverages videos generated by text-to-video models as motion guidance for animating rigged 3D characters. Rendering a given articulated 3D model into multi-view images,

AnimaX (Huang et al., 2025) trains a multi-view video-pose diffusion model that generates text-guided multi-view videos and further reconstructs 3D motion sequences. Articulate3D (Deb et al., 2025) similarly leverages multi-view rendered images, employing a self-attention rewiring mechanism to generate text relevant target poses and achieving multi-view mesh optimization through keypoint detection and alignment. These methods based on generative priors all estimate the pose of the rigged object without playing a role in the rigging process. Therefore, we first propose to use generative priors to guide the learning of skinning weights to overcome the difficulties of rigging caused by insufficient 3D data.

## 3 PRELIMINARY

**3D Gaussian Splatting.** The 3D Gaussian representation (Kerbl et al., 2023a) is a high-quality rendering model. It consists of a collection of 3D Gaussian distributions with attributes, where each Gaussian distribution $G_i$ includes: center position $\mu_i$; covariance matrix $\mathbf{\Sigma}_i$; opacity $\sigma_i$; and spherical harmonic coefficients $sh_i$. The covariance matrix $\mathbf{\Sigma}_i$ can be decomposed as the product of rotation and scaling for optimization, where the rotation matrix is represented by the quaternion $\mathbf{q}_i$, and the scaling matrix is represented by the 3D vector $\mathbf{s}_i$. When rendering an image from a specific viewpoint $v_i$, the 3D Gaussian distribution is projected onto the 2D plane to obtain the projected mean $\overline{\mu}_i$ and the projected covariance $\overline{\mathbf{\Sigma}}_i$. The color $\mathcal{C}(u)$ of a pixel $u$ is calculated as:

$$\mathcal{C}(u) = \sum_{i \in N} T_i \alpha_i \mathcal{SH}(sh_i, v_i), \quad \text{where} \quad T_i = \prod_{j=1}^{i-1}(1 - \alpha_j) \tag{1}$$

and $\mathcal{SH}(\cdot, \cdot)$ denotes the spherical harmonic function, and $\alpha_i$ can be calculated as:

$$\alpha_i = \sigma_i \exp\left(-\frac{1}{2}(u - \overline{\mu}_i)^T \overline{\mathbf{\Sigma}}_i^{-1}(u - \overline{\mu}_i)\right). \tag{2}$$

Thus, the 3D scene or object is parameterized as $\mathcal{G} = \{G_i : \mu_i, \mathbf{\Sigma}_i, \sigma_i, sh_i\}$.

**Skeleton Representation and LBS-based Deformation.** The skeleton is usually represented as a directed tree structure $\mathcal{B} = \{\mathcal{J}, \mathcal{A}\}$, where $\mathcal{J} = \{\mathbf{J}_b\}$ denotes the set of joints, and $\mathcal{A} = \{A_b\}$ represents the directed connection relationships between joints. Similar to Wu et al. (2023); Uzolas et al. (2024); Yao et al. (2025), we define global translation $\mathbf{t}$ and rotational transformations $\{\mathbf{R}_b\}_{\mathbf{J}_b \in \mathcal{J}}$, representing the rotations relative to their parent joints. Here, the parent of the root joint $\mathbf{J}_{\text{root}}$ is itself. Without loss of generality, we define $\mathbf{J}_1$ as the root node. Then a point $\mathbf{v}_j$ in the shape surface can be deformed via Linear Blend Skinning (LBS) (Lewis et al., 2000):

$$\widehat{\mathbf{v}}_j = \mathbf{P}_1 \left(\sum_{b=2}^{|\mathcal{J}|} \omega_{j,b} \mathbf{P}_b \overline{\mathbf{v}}_j\right) + \mathbf{t}, \tag{3}$$

where

$$\mathbf{P}_b = \mathbf{P}_{A_b}\hat{\mathbf{P}}_b, \ \ \hat{\mathbf{P}}_b = \begin{bmatrix} \mathbf{R}_b & \mathbf{J}_{A_b} - \mathbf{R}_b\mathbf{J}_{A_b} \\ \mathbf{0} & 1 \end{bmatrix}, \ \text{ and } \ \mathbf{P}_1 = \begin{bmatrix} \mathbf{R}_1 & \mathbf{J}_1 - \mathbf{R}_1\mathbf{J}_1 + \mathbf{t} \\ \mathbf{0} & 1 \end{bmatrix}. \tag{4}$$

Here $\mathbf{P}_b$ is defined recursively by its parent $\mathbf{P}_{A_b}$; $\overline{\mathbf{v}}_j$ is the homogeneous coordinate representation of $\mathbf{v}_j$; $\omega_{j,b}$ are the skinning weights. To simplify the representation, we denote the pose transformations of the skeleton as $\mathbf{P} = \{\mathbf{t}, \{\mathbf{R}_b\}_{\mathbf{J}_b \in \mathcal{J}}\}$. When $\mathbf{t} = \mathbf{0}$ and $\{\mathbf{R}_b\}$ are the identity matrices, we call it the rest pose and denote it as $\mathbf{P}_{\text{rest}}$.

## 4 PROPOSED METHOD

As illustrated in Fig. 2, given multi-view images $\mathcal{I} = \{I\}$ with calibrated camera parameters $\mathcal{C} = \{C\}$ depicting a complete static object, we propose $A^3$-GS, a new 3D Gaussian splatting-based framework for modeling an articulated object that can be rigged after reconstruction, enabling realistic character animation with high-quality rendering. First, we introduce a hybrid representation that integrates mesh and 3D Gaussians to reconstruct articulated objects and extract the skeleton (Sec. 4.1). The continuity of the mesh enhances the rendering quality of 3D Gaussian splatting during the animation process. Subsequently, we learn motion-coherent skinning weights by leveraging

Figure 2: Overview of our A$^3$-GS.

motion priors from the visual foundation model. To generate skeleton-controllable guidance, we select a drag-based image editing model (Shi et al., 2024) to produce tailored guidance for learning skinning weights (Sec. 4.2). By changing the skeleton pose and automatically constructing the drag signals, we obtain the edited image with fine-grained control. To mitigate the potential unreliability of generated images during skinning weight learning, we introduce a local rigidity regularization term and a refinement module to enhance the geometric quality of deformed shape (Sec. 4.3). These technical designs enable robust and stable learning of motion-coherent skinning weights.

## 4.1 3D RECONSTRUCTION AND SKELETON EXTRACTION

Due to the excessive discretization and parameter redundancy of the 3D Gaussian representation, maintaining continuity and fewer artifacts during editing or animation is challenging. To address this problem, inspired by Gao et al. (2024), we introduce a Mesh-Gaussian hybrid representation for animated objects. We bind the 3D Gaussians to a mesh, leveraging the mesh's continuity to optimize the deformation of the 3D Gaussians. Thanks to the current excellent 3D reconstruction methods, we first reconstruct the initial mesh by GOF (Yu et al., 2024). Let $\mathcal{M} = \{\mathcal{V}, \mathcal{F}\}$ denote the corresponding mesh. Each Gaussian $\mu_i$ can be assigned barycentric interpolation coordinates $(w_{f_0}^i, w_{f_1}^i, w_{f_2}^i)$ based on the nearest triangle $f \in \mathcal{F}$ defined by vertices $\mathbf{v}_{f_0}, \mathbf{v}_{f_1}$ and $\mathbf{v}_{f_2}$. Since 3D Gaussians may float above or below the mesh faces, we define:

$$\mu_i = (w_{f_0}^i \mathbf{v}_{f_0} + w_{f_1}^i \mathbf{v}_{f_1} + w_{f_2}^i \mathbf{v}_{f_2}) + \beta r \mathbf{n}_f, \tag{5}$$

where $r$ is the circumradius of the triangular face, $\mathbf{n}_f$ is the face normal, and $\beta \in [-0.5, 0.5]$ is the offset along the surface normal direction that will be learned as an additional Gaussian property. Using multi-view images as supervision, we first train these Gaussian properties and binding parameters on static objects referring to Gao et al. (2024).

During the deformation process, assuming that the deformed vertices and normals are $\widehat{\mathcal{V}} = \{\widehat{\mathbf{v}}_j\}$ and $\{\widehat{\mathbf{n}}_j\}$, the local transformation $\widehat{\mathbf{T}}_j \in \mathbb{R}^{3 \times 3}$ for each mesh vertex $\mathbf{v}_j$ can be optimized by

$$\widehat{\mathbf{T}}_j = \arg\min_{\widehat{\mathbf{T}}} \ \tau \|\widehat{\mathbf{n}}_j - \widehat{\mathbf{T}} \mathbf{n}_j\|^2 + \sum_{\mathbf{v}_k \in \mathcal{N}(\mathbf{v}_j)} \|(\widehat{\mathbf{v}}_k - \widehat{\mathbf{v}}_j) - \widehat{\mathbf{T}}(\mathbf{v}_k - \mathbf{v}_j)\|^2. \tag{6}$$

$\widehat{\mathbf{T}}_j$ can be decomposed into $\widehat{\mathbf{T}}_j = \widehat{\mathbf{R}}_j \widehat{\mathbf{S}}_j$, where $\widehat{\mathbf{R}}_j$ represents the rotational component and $\widehat{\mathbf{S}}_j$ represents the shearing component. $\mathcal{N}(\mathbf{v}_j)$ denotes the set of neighbors of $\mathbf{v}_j$. $\tau$ is a parameter to balance two terms, and we set $\tau = 0.3$ by default. Then, the covariance of each deformed Gaussian can be estimated by

$$\widehat{\mathbf{\Sigma}}_i = \widetilde{\mathbf{T}}_i \mathbf{\Sigma}_i \widetilde{\mathbf{T}}_i^T, \text{with } \widetilde{\mathbf{T}}_i = (w_{f_0}^i \widehat{\mathbf{R}}_{f_0} + w_{f_1}^i \widehat{\mathbf{R}}_{f_1} + w_{f_2}^i \widehat{\mathbf{R}}_{f_2})(w_{f_0}^i \widehat{\mathbf{S}}_{f_0} + w_{f_1}^i \widehat{\mathbf{S}}_{f_1} + w_{f_2}^i \widehat{\mathbf{S}}_{f_2}). \tag{7}$$

In the following, we use $\{\widehat{G}_i : \widehat{\mu}_i, \widehat{\mathbf{\Sigma}}_i, \sigma_i, sh_i\}$ to denote the deformed Gaussians. After binding the 3D Gaussians to the triangular mesh, the deformation of a 3D Gaussian can be transformed into the deformation of a mesh with fewer points, which reduces the degrees of freedom while enhancing continuity. We can also improve the local smoothness of the 3D Gaussian by optimizing the mesh deformation with smoothness constraints.

Using the corresponding mesh, we can also extract the skeleton $\mathcal{B}$ by the current state-of-the-art skeleton extraction methods, such as Guo et al. (2025); Song et al. (2025b;a); Zhang et al. (2025b), or manually constructed skeletons.

## 4.2 GUIDANCE GENERATION WITH MOTION PRIOR FROM VISUAL FOUNDATION MODEL

Since 3D datasets with ground-truth skinning weights or motion sequences are scarce, while 2D vision foundation models trained on large-scale data have demonstrated remarkable performance (e.g., AI (2025); JiMeng AI (2025)), especially in terms of motion rationality, we aim to leverage their motion priors to facilitate the learning of skinning weights.

A direct approach involves employing image-to-video generation models to synthesize novel motion sequences for static objects to train. However, these established models can only produce single-view videos with varying quality, and achieving consistent multi-view generation remains an open research challenge. Furthermore, even after obtaining high-quality multi-view generated videos, we must align skeletal poses with the visual content before learning skinning weights. Otherwise, there will be a misalignment between the skeleton pose and the shape pose. These significantly increase the challenges in this technical approach. Alternatively, another technical approach is to use the image-to-video model to calculate SDS loss. Yet due to the inherent unpredictability of generative models, accurately learning skinning weights without fine-grained control signals is extremely challenging.

We observe that fine-grained image editing approaches, particularly drag-based models, exhibit a remarkable similarity to skeleton-driven character animation. We adopt LightningDrag (Shi et al., 2024) as our base model. While it relies on manually specified drag, target points and editing mask, our setting demands their automatic generation to enable self-supervised training. We therefore analyze the applicability of drag-based image editing models, introduce an automatic control signal generation scheme, and demonstrate how these signals assist in learning skinning weights.

**Construct drag signals.** Since the skeleton of the static object is already available, we can generate new poses by assigning updated pose parameters. The original and transformed positions of the skeletal joints are then used as drag points and target points, respectively, to guide the generation of posed images. LightningDrag is trained with a relatively small number of drag points ($\leq 20$). Our testing and observations show that more control points lead to a significant performance drop. Therefore, we only move one joint at a time, assigning it different rotation parameters. Specifically, we first select an input image $I_l \in \mathcal{I}$ with camera parameter $C_l$. Then for each joint $\mathbf{J}_b \in \mathcal{J} \setminus \{\mathbf{J}_{\text{root}}\}$, we construct the rotation matrix $\mathbf{R}_b^{(l,\theta)}$ via axis-angle representation, setting the rotation axis to be perpendicular to the presentation plane and the uniformly sampling rotation angle $\theta$. This ensures that the joint transformations seen from the projection perspective are complete and comprehensive. Then we construct a group of pose parameters $\mathbf{P}_{(b,l,\theta)}$ by keeping the associated rotations of other joints as identity matrices. Based on the camera parameter $C_l$, we project the skeletal joints of both the rest pose $\mathbf{P}_{\text{rest}}$ and $\mathbf{P}_{(b,l,\theta)}$ onto a 2D plane, yielding projected point sets $\{\overline{J}\}_{\mathbf{P}_{\text{rest}}}$ and $\{\overline{J}\}_{\mathbf{P}_{(b,l,\theta)}}$ respectively. Points exhibiting positional displacements between $\{\overline{J}\}_{\mathbf{P}_{\text{rest}}}$ and $\{\overline{J}\}_{\mathbf{P}_{(b,l,\theta)}}$ are then identified and designated as the drag points and target points for pose $\mathbf{P}_{(b,l,\theta)}$. $\mathcal{P} = \{\mathbf{P}_{(b,l,\theta)}\}_{b,l,\theta}$ constitutes the set of candidate skeleton poses that we used for guided image generation.

**Design editing mask.** The editing mask is crucial for distinguishing areas that need to be edited and areas that remain unchanged. To this end, we compute coarse skinning weights based on the distance between the shape surface and the skeleton, which are further used to estimate the editing mask. For each bone $b$ between joint $\mathbf{J}_b$ and its parent $\mathbf{J}_{A_b}$, we compute the Euclidean distance $D_{\mathbf{v}_j,b}$ from each mesh vertex $\mathbf{v}_j$ to bone $b$. Then the initial skinning weights can be calculated as (Dionne & de Lasa, 2013):

$$\omega_{j,b}^{\text{dist}} = \left( \frac{1}{(1-\gamma)D_{\mathbf{v}_j,b} + \gamma D_{\mathbf{v}_j,b}^2} \right)^2, \tag{8}$$

where $\gamma \in [0,1]$ is a weight to control bind smoothness and we set $\gamma = 0.5$ by default. According to Eq. (3), we can calculate the deformed vertices $\widehat{\mathcal{V}}_{(b,l,\theta)}^{\text{dist}}$ with pose $\mathbf{P}_{(b,l,\theta)}$. We define the points with different positions between $\widehat{\mathcal{V}}_{(b,l,\theta)}^{\text{dist}}$ and $\mathcal{V}$ as the moving points. We set the colors of Gaussians associated with the moving points to white and the others to black, render mask images for $\{G\}$ and $\{\widehat{G}_{(b,l,\theta)}^{\text{dist}}\}$ respectively, and take their union as the editing mask $M_{(b,l,\theta)}$.

These drag points, target points, and editing mask will be fed into the drag-based image editing model to generate new images $\{\widehat{I}_{(b,l,\theta)}^{\text{drag}}\}$ for guidance. Fig. 3 shows a schematic diagram.

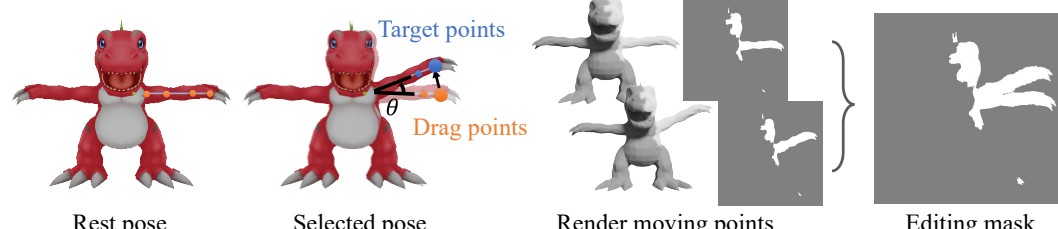

Figure 3: Construction of drag points, target points, and editing mask.

## 4.3 LEARNING OF MOTION-COHERENT SKINNING WEIGHTS

**Motion prior guided skinning learning.** Following (Zhao et al., 2022), we adopt a CNN-based neural network with explicit volumetric features to model skinning weights. We denote it as $F_\Phi(\cdot)$ parameterized with $\Phi$. Compared with modeling directly using an MLP, it exhibits stronger locality. During training, we randomly select a subset $\{\mathbf{P}_{(b,l,\theta)}\}_{(b,l,\theta)\in\mathcal{S}} \subset \mathcal{P}$. For each $\mathbf{P}_{(b,l,\theta)}$, along with the guidance image $\hat{I}_{(b,l,\theta)}$ and deformed mesh vertices $\widehat{\mathcal{V}}_{(b,l,\theta)}$, we compute the following loss function:

$$L_{(b,l,\theta)} = mL_{\text{render}}(\hat{I}_{(b,l,\theta)}, \hat{I}^{\text{drag}}_{(b,l,\theta)}) + w_{\text{g}}(1-m)L_{\text{geo}}(\widehat{\mathcal{V}}_{(b,l,\theta)}, \widehat{\mathcal{V}}^{\text{dist}}_{(b,l,\theta)}) + w_{\text{a}}L_{\text{arap}}(\widehat{\mathcal{V}}_{(b,l,\theta)}, \mathcal{V}), \quad (9)$$

where $L_{\text{render}}$ is a combination of $\ell_1$ loss and D-SSIM loss, as detailed in (Kerbl et al., 2023a). Since LightningDrag often remains static for subtle drags, we introduce $L_{\text{geo}}$ to leverage geometric information for guiding the deformation of small joints, such as fingers, that are not well captured in the generated images. In this case, we set $m = 0$, otherwise $m = 1$. See Appendix A.1 for details. Specifically, $L_{\text{geo}}$ is defined as

$$L_{\text{geo}} = \frac{1}{|\mathcal{V}_{\text{move}}|} \sum_{\mathbf{v}\in\mathcal{V}_{\text{move}}} \|\widehat{\mathbf{v}}_{(b,l,\theta)} - \widehat{\mathbf{v}}^{\text{dist}}_{(b,l,\theta)}\|^2, \quad \text{where } \mathcal{V}_{\text{move}} = \{\mathbf{v}\in\mathcal{V} | \|\mathbf{v} - \widehat{\mathbf{v}}^{\text{dist}}_{(b,l,\theta)}\| > \epsilon\}. \quad (10)$$

Here $\epsilon = 0.01$ is a threshold to select the moving points. Such a loss design encourages greater attention to detail deformation. Furthermore, since skinning weights are highly sensitive to subtle imperfections, we take advantage of the Mesh-Gaussian hybrid representation to exploit the continuity of the mesh surface and apply an as-rigid-as-possible loss:

$$L_{\text{arap}} = \sum_{\mathbf{v}_j\in\mathcal{V}} \sum_{\mathbf{v}_k\in\mathcal{N}(\mathbf{v}_j)} \|(\widehat{\mathbf{v}}_k - \widehat{\mathbf{v}}_j) - \overline{\mathbf{R}}_j(\mathbf{v}_k - \mathbf{v}_j)\|^2, \quad (11)$$

where $\overline{\mathbf{R}}_j \in \text{so(3)}$ is a rotation matrix. $w_{\text{g}}$ and $w_{\text{a}}$ are weights to balance three terms. We denote the learned skinning weights as $\{\omega^{\text{learn}}_{j,b}\}$ and the deformed mesh vertices as $\widehat{\mathcal{V}}^{\text{learn}} = \{\mathbf{v}^{\text{learn}}_j\}$.

**Local rigidity-aware refinement.** Due to the pose restrictions of LightningDrag, trained skinning weights may be discontinuous when performing coordinated actions of multiple joints. Therefore, we introduce a local rigid-aware refinement module to solve this problem. In each iteration, instead of deforming just one joint, we randomly select 1/4 of the joints except for the root joint and assign them random rotations (see the right figure).

Based on the obtained pose parameters $\mathbf{P}_r$, we deform the mesh vertices according to Eq. (3) and optimize the following loss function:

$$L_r = L_{\text{align}}(\widehat{\mathcal{V}}^r, \widehat{\mathcal{V}}^{\text{learn}}) + w^r_{\text{a}}L_{\text{arap}}(\widehat{\mathcal{V}}^r, \mathcal{V}),$$

$$\text{where } L_{\text{align}} = \frac{1}{|\mathcal{V}|} \sum_{\mathbf{v}_j\in\mathcal{V}} \|\widehat{\mathbf{v}}^r_j - \widehat{\mathbf{v}}^{\text{learn}}_j\| \quad (12)$$

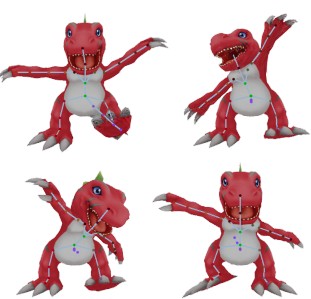

is to make the refined skinning weights and the coarse learned weights have close deformation results. $L_{\text{arap}}$ is the same as Eq. (11). Through such a simple and fast refinement module, the local smoothness of the skinning weight can be significantly improved, thereby reducing the rendering artifacts of the deformed Gaussian.

## 5 EXPERIMENTS

**Implementation Details.** We selected four viewpoints (front, back, left, and right) to generate the guidance images. For each viewpoint, we generated 4 poses for each joint with rotation angles from $-0.35\pi$ to $0.35\pi$. In the training skinning weights with motion prior, we set the number of iterations to 1500 and the parameters $w_g = 10, w_a = 10^3$. For the refinement, we set the number of iterations to 2000 and set $w_a^r = 100$ by default. These training processes both use the ADAM optimizer (Diederik, 2014). All experiments were performed on an NVIDIA RTX A6000 GPU.

**Datasets.** The test set Articulation-XL 2.0 processed by Song et al. (2025b) lacks color and ground-truth motion sequences, and some corresponding raw data from Objaverse-XL (Deitke et al., 2023b;a) also do not contain such information. Therefore, we adopted the test data processed by Zhang et al. (2025a) and sourced from Objaverse (Deitke et al., 2023b). To ensure fairness, we avoided cases from the training portion of Articulation-XL 2.0. The final test data consists of 61 objects (44 humanoids and 17 others). Each object is adjusted to its rest pose and rendered into 100 camera-view images as input. The motion sequence includes 24 frames, with corresponding meshes and rendered images from 6 camera views are used for evaluation. All images are rendered at a resolution of $512 \times 512$.

For evaluation, we rigged each object at its rest pose, fitted the skeleton poses to the ground-truth motion, and measured RMSE (Geo-err) between the deformed and ground-truth meshes, as well as the rendering discrepancies (PSNR, SSIM and LPIPS). More details can be found in Appendix A.2.

**Comparisons with State-of-the-Art Methods.** To validate the effectiveness of our learned skinning weights, we compare with state-of-the-art methods, including two template-free methods (UniRig (Zhang et al., 2025b) and Puppeteer (Song et al., 2025a)), and a humanoid-specific method (MIA (Guo et al., 2025)). For fairness, all baseline methods were applied to our reconstructed meshes for skeleton extraction and skinning weight prediction, and rendering was performed using the Mesh–Gaussian hybrid representation. Since our framework does not include a separate skeleton prediction module, we adopted the skeletons provided by the baselines to predict skinning weights. We further tested with skeletons predicted by MagicArticulate (Song et al., 2025b), which yield the best results; however, its skinning weight prediction module is not publicly available. Table 1 reports the quantitative results, and Fig. 4 presents the visual comparisons. As shown, our learned skinning weights achieve higher accuracy and smoother deformations, resulting in fewer artifacts. More visualization results can be found in Appendix A.3 and ***Supplementary Video***.

Table 1: Comparisons of the average PSNR, SSIM, LPIPS and Geo-err ($\times 10^{-2}$). "↑" (resp. ↓) indicates the larger (resp. smaller), the better. The best results are highlighted in bold.

| Dataset | Method | PSNR ↑ | SSIM ↑ | LPIPS ↓ | Geo-err ↓ |
|---|---|---|---|---|---|
| Humanoid | MIA (Guo et al., 2025) | 22.15 | 0.920 | 0.072 | 3.23 |
| | Ours (MIA) | 22.05 | 0.922 | 0.064 | 2.80 |
| | UniRig (Zhang et al., 2025b) | 21.22 | 0.911 | 0.078 | 3.18 |
| | Ours (UniRig) | 22.40 | 0.925 | **0.062** | 2.54 |
| | Puppeteer (Song et al., 2025a) | 21.50 | 0.918 | 0.070 | 2.85 |
| | Ours (Puppeteer) | 21.90 | 0.920 | 0.065 | 2.77 |
| | Ours (MagicArticulate) | **22.51** | **0.926** | **0.062** | **2.48** |
| Others | UniRig (Zhang et al., 2025b) | 23.86 | 0.947 | 0.044 | 2.32 |
| | Ours (UniRig) | 24.12 | 0.951 | 0.040 | 2.14 |
| | Puppeteer (Song et al., 2025a) | 24.04 | 0.949 | 0.043 | 2.17 |
| | Ours (Puppeteer) | 24.38 | 0.953 | 0.040 | 2.08 |
| | Ours (MagicArticulate) | **25.00** | **0.959** | **0.036** | **1.90** |

**Comparisons on Generated or Real-Captured Data.** We further evaluated the proposed method on both synthetic 3D data generated by TRELLIS (Xiang et al., 2025) and real-world captured data. For the generated 3D shape, we first obtained a complete 3D mesh and 3D Gaussian representation from a single input image using TRELLIS. Multi-view images were then rendered from 3D Gaussians and provided as input to our framework for processing. For real-captured data, we acquired dense images of a physical object using a mobile phone. The object was segmented with SAM2 (Ravi et al., 2025), and camera parameters were estimated via COLMAP (Schönberger &

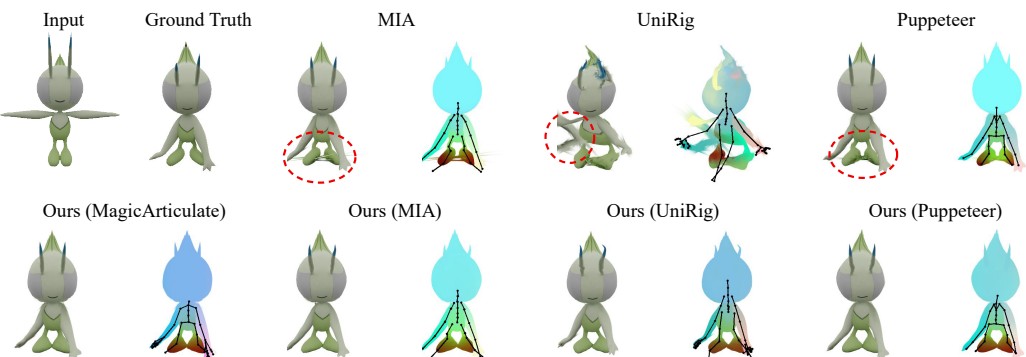

Figure 4: Visual comparison with the state-of-the-art methods.

Frahm, 2016; Schönberger et al., 2016). These preprocessed data were subsequently fed into our pipeline. We compared the proposed method with MIA (Guo et al., 2025) and show the visual results in Fig. 5 and ***Supplementary Video***. From them, we can observe that our method produces fewer artifacts during the animation process, demonstrating the practicality of our approach across more data sources.

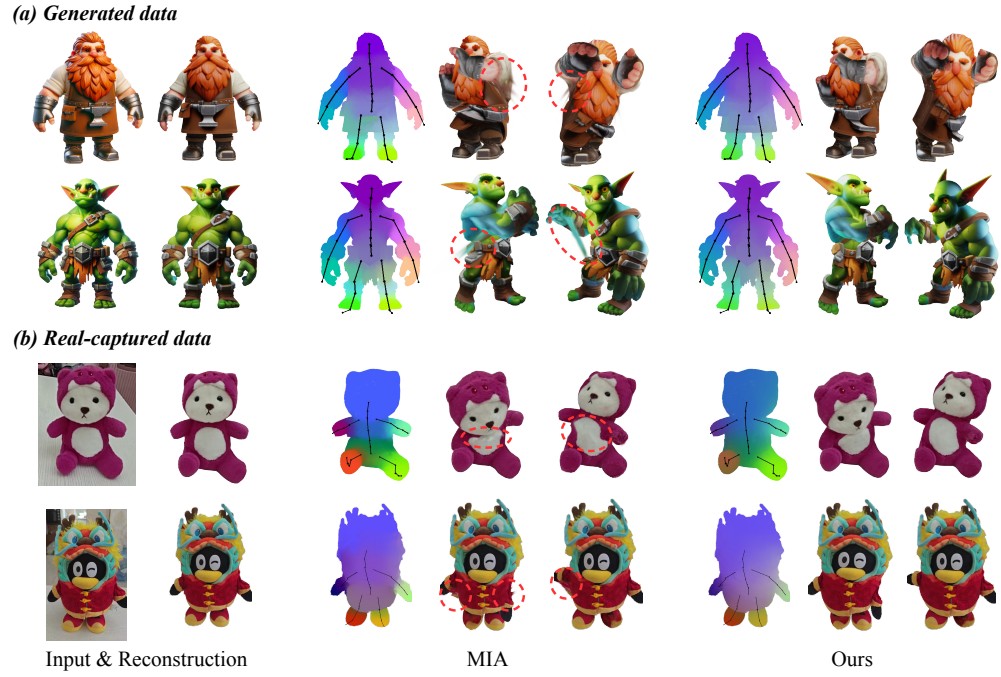

Figure 5: Visual comparisons of our method on generated 3D data and real-captured data. The figure shows: input image (one of the real data), the rendering image of the reconstructed model, the learned skinning and two novel poses by MIA and our method.

**Ablation Studies.** To verify the effectiveness of the Mesh-Gaussian hybrid representation, we first compared it with the original 3D Gaussian representation (*Aniso-G*), similar to those employed in (Wan et al., 2024; Yao et al., 2025). Yao et al. (2025) mentioned that setting the Gaussians to be isotropic can help reduce artifacts, so we also tested this setting (*Iso-G*). Due to the high discretization and lack of local continuity in 3D Gaussians, they are prone to artifacts such as burrs and tears, as shown in Fig. 8. To further validate the necessity of the mesh component, we tested an alternative approach that replaces the mesh with downsampled Gaussian points and substitutes the mesh connectivity with local k-neighborhoods, inspired by ARAP-GS (Han et al., 2025) for Gaussian editing (*Sampled-G*). From the comparative results in Table 2, it is evident that the Mesh-

Table 2: Comparisons of different variants for deformable Gaussian representation.

| Variant | PSNR$^\uparrow$ | SSIM$^\uparrow$ | LPIPS$^\downarrow$ |
|---|---|---|---|
| Iso-G | 22.08 | 0.921 | 0.071 |
| Aniso-G | 22.34 | 0.924 | 0.068 |
| Sampled-G | 22.80 | 0.930 | 0.064 |
| Ours | 23.25 | 0.936 | 0.054 |

*Due to out-of-memory errors for Iso-G and Aniso-G on one case, we report the statistical results on the remaining cases.

Table 3: Comparisons of the average precision (Geo-err ($\times 10^{-2}$)) with different variants.

| Variant | PSNR$^\uparrow$ | SSIM$^\uparrow$ | LPIPS$^\downarrow$ | Geo-err$^\downarrow$ |
|---|---|---|---|---|
| init skin. | 22.48 | 0.927 | 0.064 | 2.66 |
| w/o $L_{\mathrm{arap}}$ | 22.84 | 0.932 | 0.058 | 2.54 |
| w/o $L_{\mathrm{geo}}$ | 23.22 | 0.936 | 0.054 | 2.45 |
| w/o refine. | 23.16 | 0.935 | 0.055 | 4.02 |
| Ours | 23.20 | 0.936 | 0.055 | 2.32 |

Gaussian hybrid representation we used significantly enhances the rendering quality. Fig. 8 (see Appendix) in the appendix demonstrates that our method significantly reduces artifacts, leading to superior rendering quality.

To verify the effectiveness of each component of our method, we first tested using the initial skinning weights in Eq. (8) directly without training. As shown in the Table 3, it leads to a significant drop in accuracy. We also tested the impact of different loss terms on our approach during motion prior guided skinning learning process. We ignored $L_{\mathrm{geo}}$ and $L_{\mathrm{arap}}$ repectively. From Table 3, it can be observed that the use of the $L_{\mathrm{arap}}$ loss significantly enhances the accuracy of the model. Although the precision of the variant without $L_{\mathrm{geo}}$ appears relatively high, the visual results provided in Fig. 6 demonstrate that it leads to noticeable misalignments in fine-grained joints (fingers) and shape. Furthermore, the variant without refinement exhibit increased geometric error due to pronounced local surface discontinuities. All experiments in this part are tested on all test data using the skeleton extracted by MagicArticulate (Song et al., 2025b). Additional results are provided in Appendix A.3.

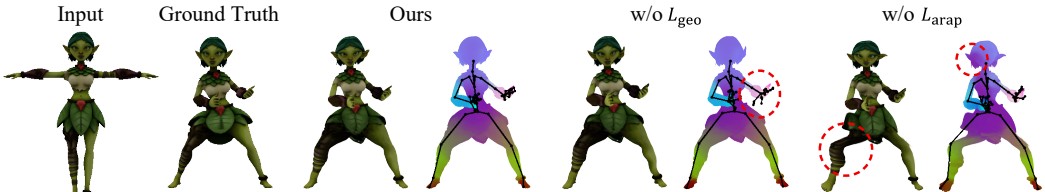

Figure 6: Visual comparison for different variants on loss terms.

## 6 CONCLUSION AND DISCUSSION

We proposed a new framework for reconstructing articulated objects from multi-view images, extracting their skeletal structures, and learning skinning weights to achieve realistic animation and high-quality rendering. While 3D Gaussians offer efficient and high-fidelity rendering capabilities, their highly discrete nature limits performance in animation tasks. To address this problem, we introduced a Mesh-Gaussian hybrid representation for modeling articulated objects, leveraging the continuity of the mesh to estimate deformed Gaussians and reduce artifacts during animation. Furthermore, due to the scarcity of 3D data with realistic skeletal rigging or motion sequences, existing methods often suffer from limited generalization. Thus, we utilized motion priors provided by visual foundation models to guide the learning of motion-coherent skinning weights. To enhance the skeleton controllability of the visual foundation model, we adopted a drag-based image editing approach and designed an automated control signal generation mechanism. Additionally, we introduced a local rigidity regularization constraint to mitigate the impact of semantically inappropriate generated images and improve the stability and robustness of skinning weight learning.

While our method performs well on many cases, its main limitation lies in the mismatch between the skeleton-driven motion and the behavior of the drag-based image editing model. For instance, dragging a head joint in skeleton space corresponds to a left–right swing, whereas the editing model may instead interpret it as turning the head left or right. Such misalignments can prevent the visual model from providing effective guidance. A promising direction for improvement is fine-tuning the drag-based image editing model with skeleton-driven data to better align the two modalities.

STATEMENT

ETHICS STATEMENT

We adhere to the IClR Code of Ethics in this research. The datasets used are publicly available with no inclusion of private, sensitive, or proprietary data involving human/animal subjects. Our work focuses on 3D reconstruction and animation of digital assets, with no foreseeable ethical risks or potential for harm.

REPRODUCIBILITY STATEMENT

We have described the method steps in detail in Sec. 4, elaborated on the implementation and evaluation details in Sec. 5 and Appendix A.1,A.2, and listed the UIDs in Objaverse dataset (Deitke et al., 2023b) of the test datasets in the Supplementary Materials. All experiments were conducted on a workstation equipped with an Intel Xeon 4309Y CPU and an NVIDIA RTX A6000 GPU. Our code, along with instructions for data preprocessing, will be publicly available, thus ensuring strong reproducibility.

STATEMENT OF LLMS USAGE

We used large language models (LLMs), i.e., DeepSeek, solely for grammar polishing of the manuscript. All LLM outputs were manually verified for accuracy, and no content was directly adopted without validation. The authors bear full responsibility for all content.

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

# A APPENDIX

## A.1 TECHNICAL DETAILS

In this section, we provide additional technical details. In Sec. 4.3, we mentioned that LightningDrag will remain motionless for small edits, such as finger movements. To automatically detect such cases, we designed two detection mechanisms. Firstly, we compute an edit ratio $r_{\text{edit}}$. At the rest pose, we rendered the moving points to obtain a mask of the movable region, and then calculated the area ratio of the movable region to the object's mask as $r_{\text{edit}}$, as shown in Fig. 7. When $r_{\text{edit}} < 0.1$, we assume that the generated images will be unreliable and set $m = 0$ in Eq. (9). Secondly, we check the similarity between the generated image and the input reference image with the rest pose. When their $\ell_1$ loss is less than $0.015$, we consider the generated image to be unchanged and set $m = 0$.

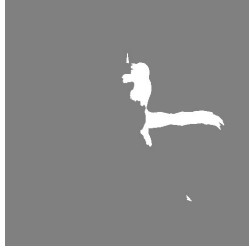 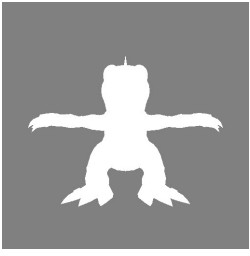

Movable mask        Object's mask

Figure 7: Example of the movable mask and object's mask used for calculating the edit ratio.

For the running time, the reconstruction phase takes approximately 40 minutes, which includes 30 minutes for mesh reconstruction using GOF (Yu et al., 2024) and under 8 minutes for training the Mesh-Gaussian hybrid representation. The rigging phase requires about 50 minutes in total, including a few seconds for skeleton extraction, 8 minutes for generating the guided images, around 40 minutes for motion-pair-guided skinning learning, and 2 minutes for local rigidity-aware refinement.

## A.2 EVALUATION DETAILS

Since the comparison methods extract skeletons with inconsistent numbers of joints and different topologies for each object, it is not feasible to directly compare the quality of their predicted skinning weights. However, as our test cases provide ground-truth motion sequences, we optimized the skeletal poses of each method to fit the ground-truth motion sequences and then measured the discrepancy between the fitting results and the ground truth. To improve the fitting quality and temporal continuity of the optimized pose, we employed an 8-layer MLP with 256-dimensional features to model the poses, and optimized them for each object by solving the problem:

$$\mathbf{P}^* = \arg\min_{\mathbf{P}} \sum_{\mathbf{v}_j \in \mathcal{V}} \|\widehat{\mathbf{v}}_j - \mathbf{u}_j\|^2, \tag{13}$$

where $\widehat{\mathbf{v}}_j$ denotes the deformed position of $\mathbf{v}_j$ obtained by LBS-based deformation in Eq. (3) using the existing methods or our method. $\mathbf{u}_j$ is the ground-truth corresponding point of $\widehat{\mathbf{v}}_j$, which can be obtained by nearest neighbor searching between the reconstructed mesh and the ground-truth mesh at the rest pose. The ADAM optimizer (Diederik, 2014) was adopted during the optimization and the number of iterations is set $5000$. It is worth noting that during this process, the skeleton and learned skin weights remain fixed. The deformed position $\{\widehat{\mathbf{v}}_j^*\}$ with optimized pose $\mathbf{P}^*$ can be used to compute the geometric error

$$\text{Geo-err}(\{\widehat{\mathbf{v}}^*\}) = \sqrt{\sum_{\mathbf{v}_j \in \mathcal{V}} \|\widehat{\mathbf{v}}_j^* - \mathbf{u}_j\|^2 / |\mathcal{V}|}. \tag{14}$$

$\{\widehat{\mathbf{v}}_j^*\}$ are also used to deform Gaussians according to Eqs. (5),(7) for multi-view rendering and calculation of rendering error.

### A.3 MORE VISUALIZATION RESULTS AND ANALYSIS

**Visualization Results of Ablation Studies.**    Fig. 8 shows the results obtained from different deformable Gaussian representations. From the figure, we can observe that the original 3D Gaussian representation (*Aniso-G*) and Gaussians with downsampled control points for deformation (*Sampled-G*) produce more flickering artifacts. While using isotropic 3D Gaussians (*Iso-G*) reduces flickering artifacts, it introduces noticeable holes. Our proposed Mesh-Gaussian hybrid representation significantly mitigates these rendering artifacts.    We also show the visual results with the initial skinning weights in Eq. (8) and without the refinement module in Fig. 9. We can see that the quality of initial skin weights is quite poor, and the refinement module improves the smoothness of the multi-joint coordinated motion.

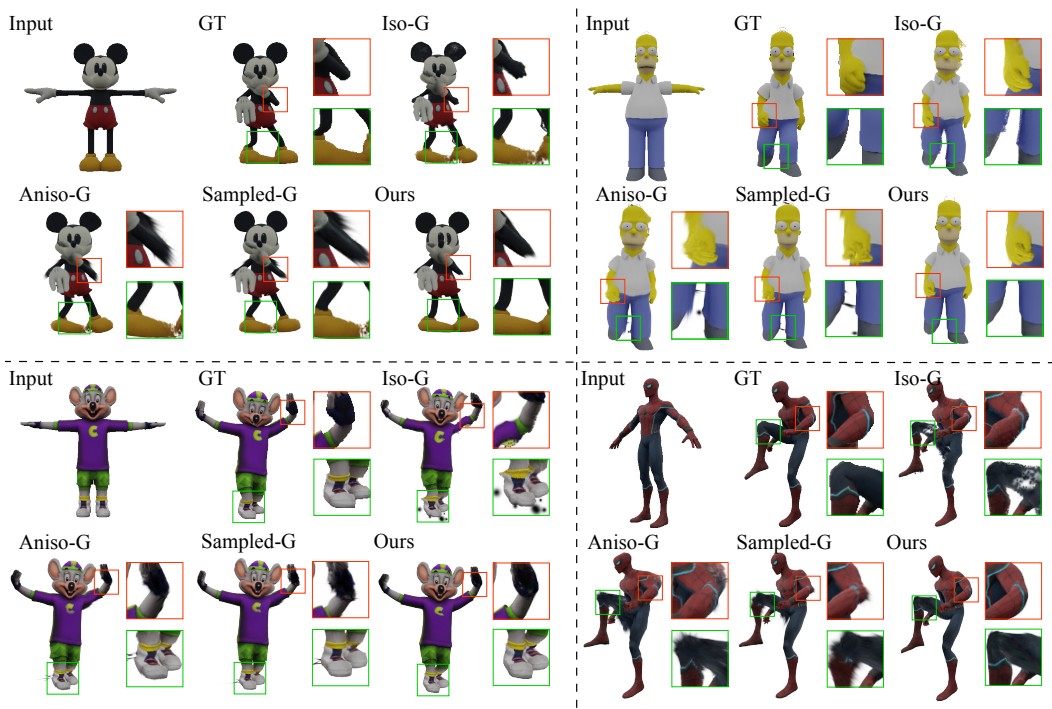

Figure 8: Visualization results of different variants of deformable Gaussian representation.

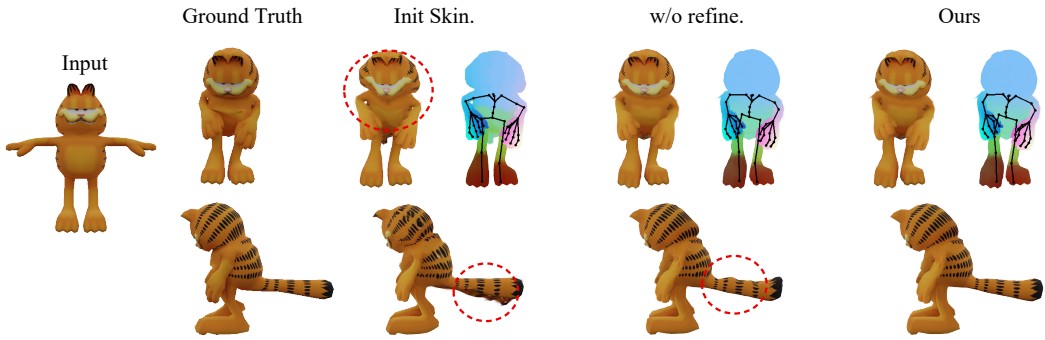

Figure 9: Visual comparison for different variants.

**Dependency analysis of skeleton.**    Although our method relies on existing skeleton extraction techniques to obtain the skeleton, our skinning prediction exhibits a certain degree of tolerance to skeleton quality. As shown in Fig. 10, even when the skeleton contains slightly inaccurate offsets (Fig. 10 (1)) or redundant bones (Fig. 10 (2)), we can still obtain reasonable skinning weights. Our method can also predict reasonable skinning weights for partially accurate skeletons, see (3) and

(4) in Fig. 10. However, when the skeleton is completely wrong (Fig. 10 (5)), we cannot handle it. Since skeletons are sparse and easier to acquire compared to dense skinning weights, models predicting skeletons are more likely to achieve good performance. By leveraging visual model priors to learn skinning weights, our approach can facilitate the creation of more datasets. Our framework is expected to drive the development of large-scale models for 3D Gaussian rigging.

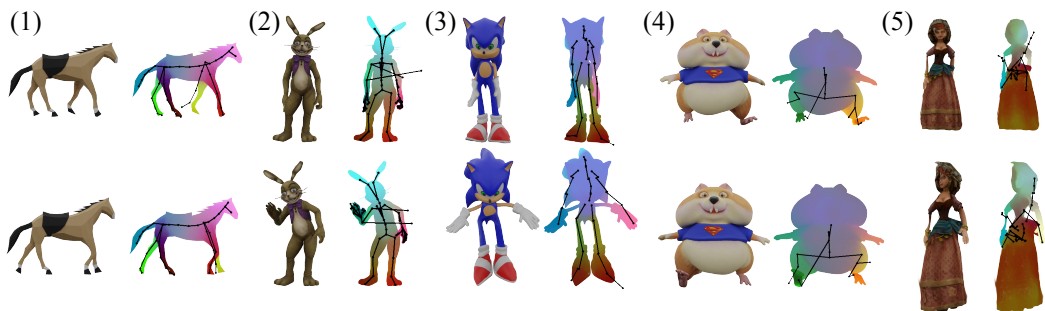

Figure 10: Our method under imperfect skeletons. For each object, we visualized two actions and their skeleton and skinning weights.

**Robustness Analysis of Motion Priors.** We obtain motion priors from the LightningDrag model and have also tested its robustness regarding the number of drag points and the magnitude of movement. We conducted experiments with single-joint rotations, as well as rotations of 10%, 30%, and 50% of the total joints. For each case, we varied the rotation angle from $0.1\pi$ to $0.7\pi$. The test results are shown in Figures 12 and 13. From these figures, we can observe that both a larger number of drag points and a greater range of motion lead to a degradation in the quality of the generated images. Therefore, we employ single-joint rotation to construct the control signal for image generation and use relatively small rotation angles ($< 0.35\pi$) to reduce artifacts in the generated images, thereby enhancing training stability. Our experimental results demonstrate that this setup obtains good skinning results and achieves superior performance compared to existing approaches.

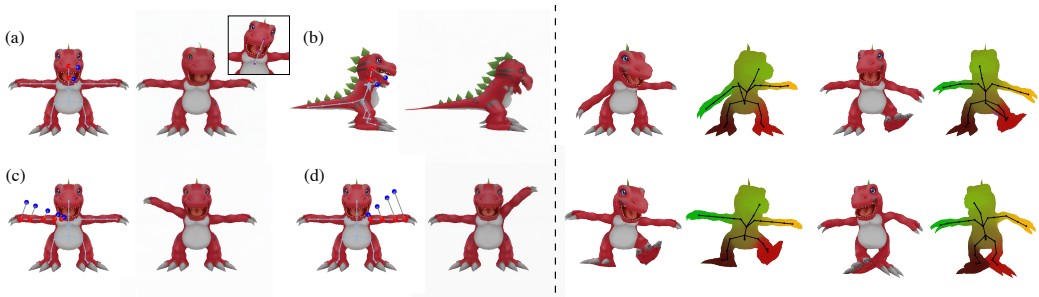

Figure 11: The figures on the left show the drag signals and the generated images, while the figures on the right present the animation results derived from the learned skinning weights. Specifically, (a) illustrates the semantic mismatch between the generated image induced by drag signals and the skeleton-based deformation, where the top-right corner shows an expected posture. The red and blue dots denote drag points and target points, respectively.

We have also analyzed the impact of the subtle misalignment between drag-based image editing and our specific task on the results. Since drag-based image editing models typically interpret the movement of frontal facial points as turning the head left or right rather than a left-right swing, this may lead to a semantic mismatch with our intended motion, as illustrated in Fig. 11 (a). However, from a side view, the problem is partially alleviated (Fig. 11 (b)). Moreover, when other parts such as the arms move, the head maintains a stable and reasonable pose (Fig. 11 (c) (d)). Therefore, such information helps improve the training of head skinning weights, resulting in a minimally noticeable effect.

Although we rely on the LightningDrag model to learn skinning weights, it is important to emphasize that our framework can flexibly incorporate any better drag-based image editing model. With the rapid development of visual foundation models, image editing models are expected to achieve better performance. Current limitations will thus be mitigated, and the performance of our method will consequently improve.

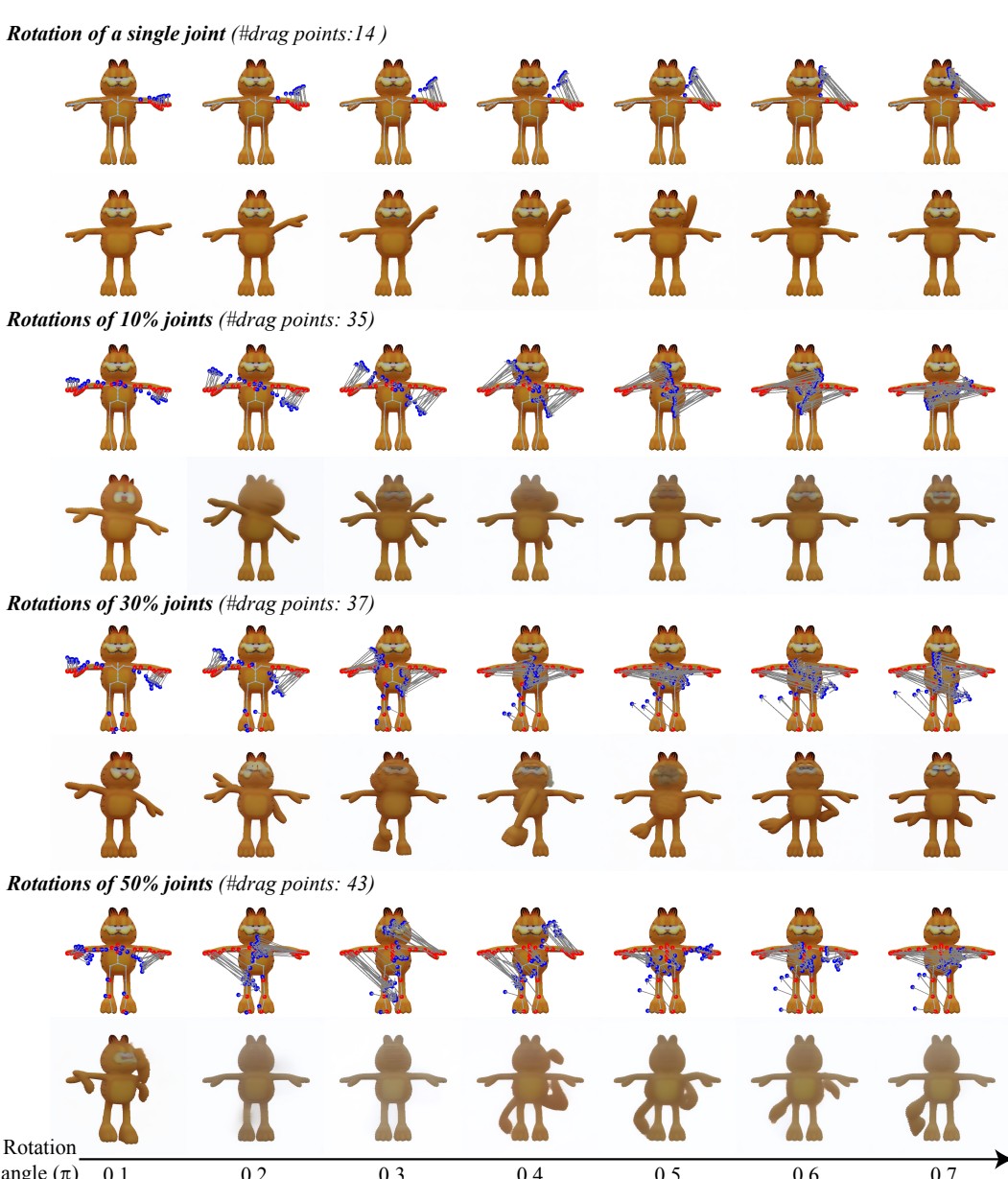

Figure 12: Performance testing of different drag signals for LightningDrag. The red and blue dots denote drag points and target points, respectively.

**More Visual Comparisons.** We also show more visual comparisons with the state-of-the-art methods on humanoid shapes (Fig. 14) and other shapes (Fig. 15). From these figures, we can observe that the proposed method produces higher quality skinning weights and resulting smooth deformations lead to very few rendering artifacts.

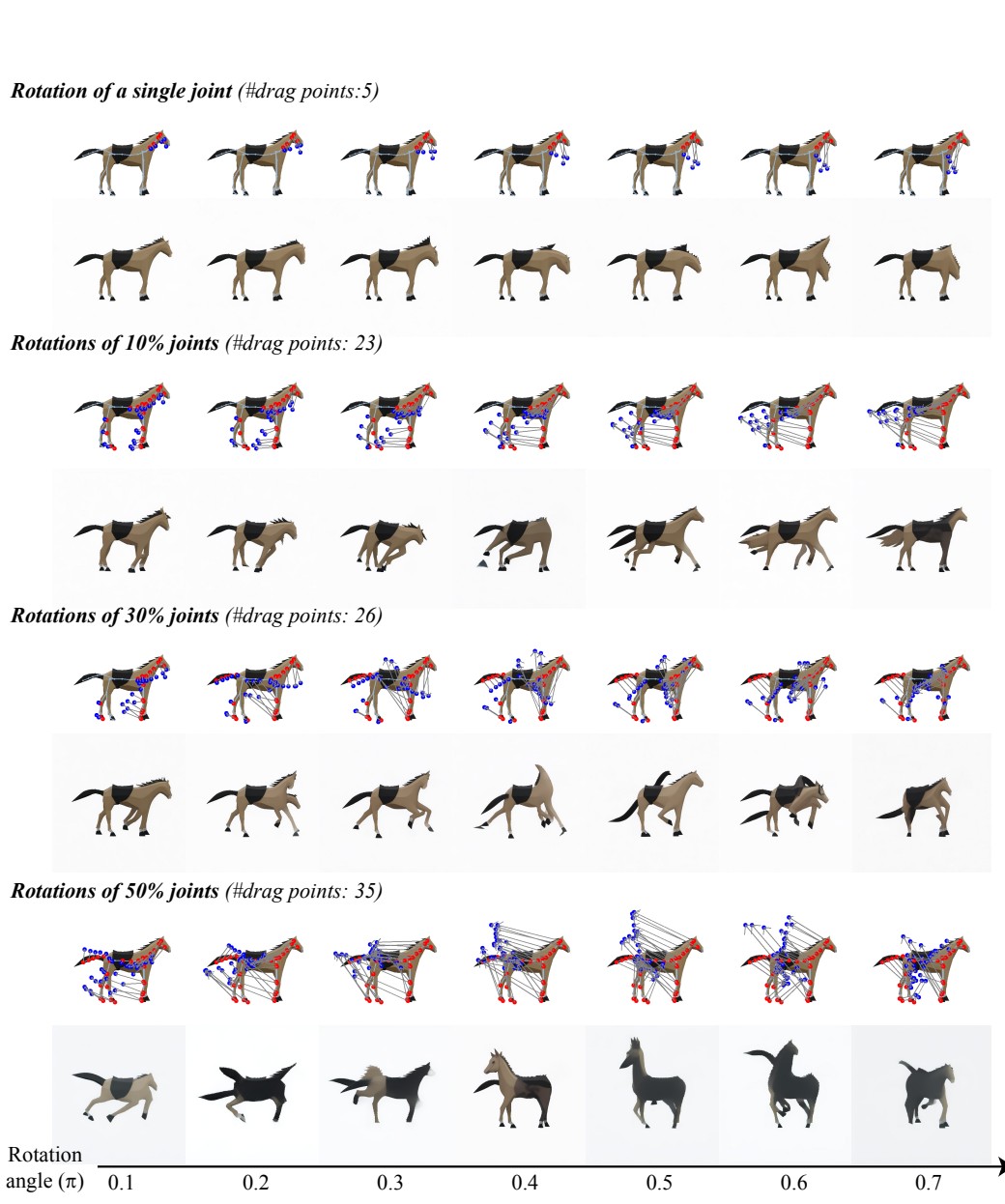

Figure 13: Performance testing of different drag signals for LightningDrag. The red and blue dots denote drag points and target points, respectively.

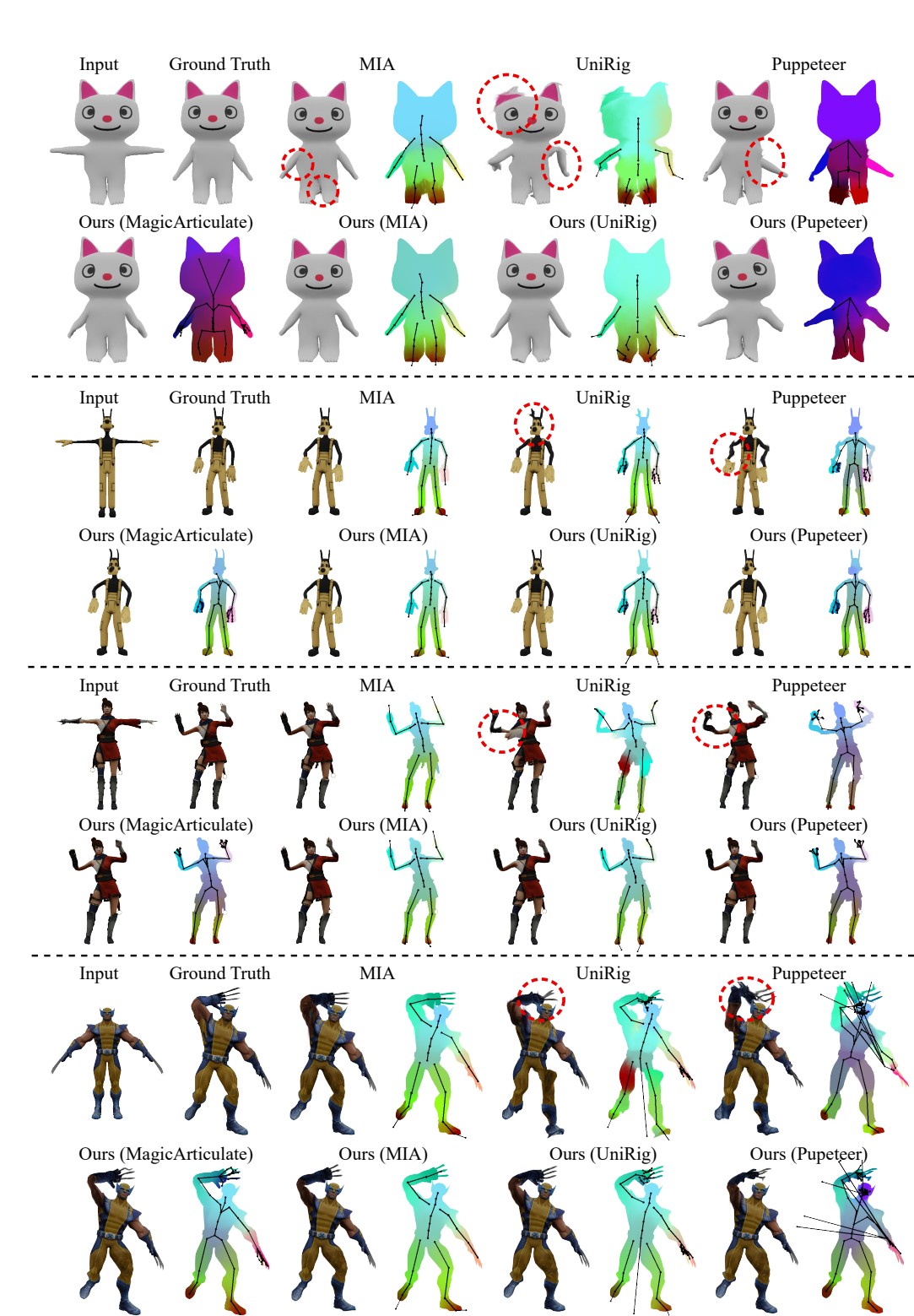

Figure 14: Visual comparison with the state-of-the-art methods on humanoid shapes.

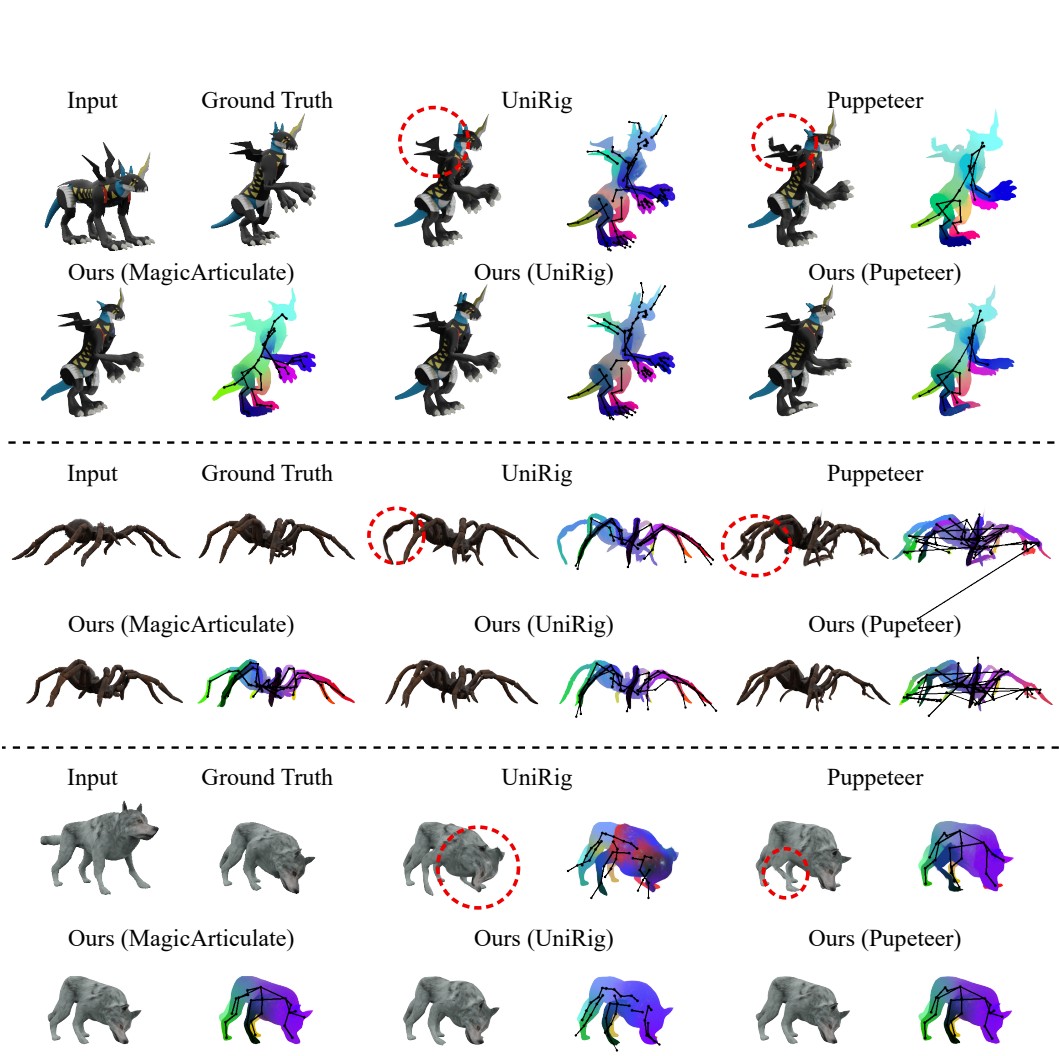

Figure 15: Visual comparison with the state-of-the-art methods on other shapes.

