# OpenReview forum: "A$^3$-GS: Animate Any Articulated Objects with 3D Gaussian Splatting"
_ICLR.cc/2026/Conference — Submitted to ICLR 2026_

### Official Review · Reviewer_1iVk · 2025-10-17

**Soundness:** 4
**Presentation:** 3
**Contribution:** 3
**Rating:** 6
**Confidence:** 4

**Summary:**

This paper proposes A³-GS, a framework for reconstructing articulated objects from multi-view images and automatically rigging them for animation. The key contributions include: (1) a Mesh-Gaussian hybrid representation that binds 3D Gaussians to mesh surfaces to reduce animation artifacts, (2) leveraging motion priors from a drag-based image editing model (LightningDrag) to learn skinning weights without extensive 3D training data, and (3) incorporating ARAP regularization and a refinement module for smoother deformations. The method demonstrates improvements over existing rigging methods.

**Strengths:**

1. Novel approach to a practical problem: Using drag-based image editing models to provide motion priors for skinning weight learning is creative and addresses the scarcity of 3D rigged datasets.

2. Mesh-Gaussian hybrid representation: The combination of mesh continuity with Gaussian splatting for animation is well-motivated and helps reduce common artifacts.

3. Comprehensive pipeline* The paper presents an end-to-end system from reconstruction to animation with reasonable technical design.

4. Experimental validation: Quantitative comparisons on multiple metrics (PSNR, SSIM, LPIPS, Geo-err) and ablation studies support the design choices.

**Weaknesses:**

1. Limited novelty and contribution in individual components: The Mesh-Gaussian binding closely follows GaussianMesh, and the skeleton extraction relies entirely on existing methods. The main novelty is in combining LightningDrag for skinning weight learning and the training strategy.

2. Missing comparisons with Gaussian-based animation methods. The paper claims superiority over discrete Gaussian methods but provides no experimental evidence: Lines 117-122 cite RigGS (Yao et al., 2025), ARAP-GS (Han et al., 2025), and other Gaussian animation methods, stating they "tend to exhibit artifacts" or "compromise rendering quality." However, Table 1 contains zero comparisons with any of these methods. All baselines (MIA, UniRig, Puppeteer) are mesh-based skinning weight predictors, not Gaussian animation methods.

**Questions:**

1. How sensitive is the method to the quality of initial skeleton extraction? What happens when the skeleton topology is incorrect?

2. The paper mentions LightningDrag is trained with ≤20 drag points and performance drops with more points. How does this limit the complexity of objects that can be rigged? Can you provide a failure case analysis?

---

> ### Author Response · Authors · 2025-11-22
> **Response to Reviewer 1iVk**
>
> ```
> 1. Limited novelty and contribution in individual components: The Mesh-Gaussian binding closely follows GaussianMesh, and the skeleton extraction relies entirely on existing methods. The main novelty is in combining LightningDrag for skinning weight learning and the training strategy.
> ```
> **Response**:
> We propose a new framework for reconstructing rigged 3D models from multi-view images of static articulated template-free objects using 3D Gaussian representation, enabling high-quality and efficient rendering and animation. Due to the discrete nature of Gaussian representations, we introduce a Mesh-Gaussian hybrid representation to reduce artifacts like burrs and tears during animation. Although this hybrid representation has been previously proposed for editing 3D Gaussian scenes, we are **the first to apply it to articulated object representation and drive animation via skeletal control**. Secondly, the scarcity of 3D rigged data limits the effectiveness of current skinning prediction methods, while manually designing skinning weights is labor-intensive. To our knowledge, we are **the first to introduce a vision-based model for learning skinning weights** of the template-free objects. To address the uncontrollability of visual generative models, we incorporate a drag-based image-editing model and design techniques such as using projected skeletal points as control signals, achieving data-free skinning learning. Our results demonstrate the effectiveness of the proposed method.
>
>
> ```
> 2. Missing comparisons with Gaussian-based animation methods. The paper claims superiority over discrete Gaussian methods but provides no experimental evidence: Lines 117-122 cite RigGS (Yao et al., 2025), ARAP-GS (Han et al., 2025), and other Gaussian animation methods, stating they "tend to exhibit artifacts" or "compromise rendering quality." However, Table 1 contains zero comparisons with any of these methods. All baselines (MIA, UniRig, Puppeteer) are mesh-based skinning weight predictors, not Gaussian animation methods.
> ```
> **Response**:
> RigGS takes monocular 2D videos as input to supplement dynamic objects, while our input is multi-view images of static objects. ARAP-GS performs 3D Gaussian scene editing, while we perform skeletal rigging and animation. Therefore, our task differs from theirs and cannot be directly compared. To verify the effectiveness of the Mesh-Gaussian hybrid representation in our framework, we added a comparison of variants in Sec. 5 of the updated PDF file by replacing our Mesh-Gaussian representation with a 3D Gaussian representation and downsampled points as control points to drive the Gaussian representation. Iso-G and Sampled-G in Table 2 are similar to the representations of RigGS and ARAP-GS, respectively. From Table 2 and Fig. 8 of the updated PDF file, we can observe that our method achieves higher render quality compared to these methods.
>
> --------
> Table 2: Comparisons of different variants for deformable Gaussian representation.
> | Variant | PSNR $^{\uparrow}$ | SSIM $^{\uparrow}$ | LPIPS $^{\downarrow}$ |
> | :---: | :---: | :---: | :---: |
> |  Iso-G | 22.08 | 0.921 | 0.071 |
> |  Aniso-G | 22.34 | 0.924 | 0.068 |
> |  Sampled-G | 22.80 | 0.930 | 0.064 |
> |  Ours | **23.25** | **0.936** | **0.054** |

---

> ### Author Response · Authors · 2025-11-22
> **Response to Reviewer 1iVk**
>
> ```
> Q-1. How sensitive is the method to the quality of initial skeleton extraction? What happens when the skeleton topology is incorrect?
> ```
> **Response**:
> We have added a discussion on the dependence of skeleton extraction in Appendix A.3 of the updated PDF file. As shown in Fig. 10, we have a certain tolerance for skeleton quality. When the skeleton contains slightly inaccurate offsets (Fig. 10 (1)) or redundant bones (Fig. 10 (2)), we can still obtain reasonable skinning weights. Our method can also predict reasonable skinning weights for partially accurate skeletons, see (3) and (4) in Fig. 10. However, when the skeleton is completely wrong (Fig. 10 (5)), we cannot handle it. However, due to its sparsity, the skeleton is easier to obtain than skinning weights, for example, through hand-drawing. This makes it easier
> to increase the skeleton data to improve the skeleton prediction model’s capabilities. Ground-truth skinning weights, on the other hand, require experienced designers to spend more time creating them. Therefore, the core module of our method provides an automated approach for estimating skinning weights, and is not constrained by the lack of ground-truth data, offering a viable solution for constructing large-scale datasets.
>
> ```
> Q-2. The paper mentions LightningDrag is trained with ≤20 drag points and performance drops with more points. How does this limit the complexity of objects that can be rigged? Can you provide a failure case analysis?
> ```
> **Response** :
> We have added an analysis of drag-based models where more joint perturbations construct more drag points in Appendix A.3 of the updated PDF file. from Figures 11 and 12, we can see that more drag points introduce instability, leading to poor generation results. Our design of single-joint rotation perturbation fully leverages the advantages of LightningDrag, effectively mitigating the impact of unreliable generated images. Combined with regularization constraints such as ARAP, our framework ensures stable training and produces robust results, outperforming existing methods. Our **Supplementary Video** presents rich and intuitive comparison results. Furthermore, our method is not limited to LightningDrag, as our framework can **flexibly adopt any better drag-based image-editing model**. With the rapid development of 2D base models, richer motion priors will significantly improve our performance.

---

> > ### Comment · Reviewer_1iVk · 2025-11-25
> >
> > Thank you for your reply; it has resolved my concerns.  Considering the contribution of this work, I maintain the original score of 6.

---

> > > ### Author Response · Authors · 2025-11-25
> > > **Response to Reviewer 1iVk**
> > >
> > > We are glad to know that all your concerns have been well resolved.  Thanks for your time and effort in reviewing our manuscript, as well as your recognition of our work and positive rating. However, we are somewhat not persuaded by your underestimation of our contributions. We believe our contributions are significant: we have established a new framework to animate objects represented by 3D Gaussians, which involves a substantial workload. Creatively leveraging motion priors from a drag-based image editing model, we learned skinning weights and ingeniously adopted a Mesh-Gaussian hybrid representation, greatly reducing artifacts during the rendering process. Our method also achieved better performance compared to existing methods and has the potential to build a large dataset for 3D Gaussian rigging.

---

> > > > ### Comment · Reviewer_1iVk · 2025-11-25
> > > >
> > > > First of all, the improvement of this work compared to the previous method is not very significant.  Secondly, it can be seen from the ablation study that L_geo and refinement have little impact on the effect. ARAP loss is a technique that has been widely verified long ago, and only "init skin" contributes. I think the 6 is a reasonable score.

---

> ### Author Response · Authors · 2025-11-25
> **Response to Reviewer 1iVk**
>
> Thank you for your detailed reply. However, **we are concerned that the numerical results in Table. 1 may have biased your assessment** . In fact, since our task is to rig a skeleton to a static object and estimate skinning weights, rather than perform dynamic reconstruction from a given video. We can only adjust the skeleton pose to best match the target motion sequence. However, due to the **lack of perfect pixel-level alignment** between the rendered video and the ground-truth video, the rendering accuracy of all methods is somewhat limited. To facilitate a clearer comparison between the methods, we provide rich visualization results in the updated **Supplementary Video** and in the Appendix (**Figures 13 and 14** in the updated PDF file). As shown, the visual quality achieved by our method is significantly superior to that of the compared approaches.
>
> Furthermore, our main contributions may still be misunderstood. One of our contributions is being **the first** to propose the use of a drag-based image editing model for learning skinning weights, introducing **a novel and comprehensive framework**. Technical designs such as the ARAP loss, $L_{\text{geo}}$, and initial skinning weights are integral components aimed at enhancing the framework's performance.
> While the refinement module yields only marginal gains in rendering accuracy metrics such as PSNR, it significantly reduces the geometric error in 3D space (from 4.02 to 2.32 in Table 3). This enhancement is attributed to the improved overall smoothness provided by this module.
> Moreover, since $L_{\text{geo}}$ and the refinement module aim to **enhance local details**, their contribution is not fully reflected in the quantitative metrics. Nevertheless, the results in **Figures 6 and 9** demonstrate that they are visually critical.

---

### Official Review · Reviewer_YFkN · 2025-10-26

**Soundness:** 3
**Presentation:** 2
**Contribution:** 3
**Rating:** 6
**Confidence:** 2

**Summary:**

The paper aims to articulate a 3D avatar from multiview images. Since the input lacks deformation information, additional priors are required to guide the articulation. To learns the articulation, it first reconstructs the 3D object from the multiview images and binds the skeletons automatically. Then this paper proposes to use the priors from drag-based diffusion models for learninig the articulation: It generates videos following the designed drag signal. The linear blending weights are optimized from the generated videos.

The key novelty is the adoption of drag-based diffusion models over image-to-video or multiview video diffusion approaches. Their precise controllability simplifies and stabilizes the learning process. The paper also proposes the automatic way to construct the control signals to the diffusion models.

It demonstrates its superior performance on diverse characters over various baselines.

**Strengths:**

* Its main contribution—the adoption of drag-based diffusion—is well motivated. The paper provides a detailed discussion of the sources of deformation priors and compares drag-based diffusion with alternative approaches, effectively highlighting its advantages.
* It describes how to genrate the control signals to drag-based diffusions in details. As this is the core part, the details help understand the method.
* It shows better qualitative and quantitative results over other baselines, demonstrating the effectiveness of the method.

**Weaknesses:**

* Although the paper is titled “Animate Any Articulated Objects,” the demonstrated examples are limited to 3D characters. Common articulable objects, such as laptops, are not included in the results. This limitation may stem from the skeleton-binding algorithms adopted, which could restrict the reconstruction and articulation of non-character objects.
* The articulation is limited to part-wise rigid transformations. Although objects may exhibit secondary motions in skeleton-driven animations (e.g., the hair dynamics in the second example of Fig. 1), the paper does not demonstrate the ability to model such effects. While drag-based diffusion models are capable of capturing these non-rigid motions in 2D, the adopted 3D representation appears to constrain the articulation to rigid transformations only.

**Questions:**

Please see the weaknesses.

---

> ### Author Response · Authors · 2025-11-22
> **Response to Reviewer YFkN**
>
> ```
> Although the paper is titled “Animate Any Articulated Objects,” the demonstrated examples are limited to 3D characters. Common articulable objects, such as laptops, are not included in the results. This limitation may stem from the skeleton-binding algorithms adopted, which could restrict the reconstruction and articulation of non-character objects.
> ```
> **Response**:
> Our work focuses on articulated objects of character type, similar to works such as MagicArticulate[Song et al. 2025b] and RigGS[Yao et al. 2025]. Unlike strictly part-wise rigid objects such as laptops, cabinets and drawers, the characters we handle are not completely part-wise rigid and exhibit non-rigid deformations in many regions, such as muscle stretching. For part-wise rigid models, they typically model fixed rotation axes, translation axes, rotation angles, and translation distances, featuring fewer joints and simpler topology with lower degrees of deformation freedom. In contrast, our deformation field involves arbitrary rotation axes at each joint and generally no translation (e.g., an arm cannot separate from the body). Therefore, the deformation models for these non-character part-wise rigid objects differ from ours. Our framework is theoretically applicable for these non-character objects, but it requires adaptations such as replacing the skeleton prediction method, slightly adjusting the skeleton-based deformation model, and modifying deformation constraints. Thus, it cannot be directly applied to these non-character objects.
>
>
> ```
> The articulation is limited to part-wise rigid transformations. Although objects may exhibit secondary motions in skeleton-driven animations (e.g., the hair dynamics in the second example of Fig. 1), the paper does not demonstrate the ability to model such effects. While drag-based diffusion models are capable of capturing these non-rigid motions in 2D, the adopted 3D representation appears to constrain the articulation to rigid transformations only.
> ```
>
> **Response**:
> Firstly, our skeleton-based model does not strictly adhere to a part-wise rigid transformation. Although each joint in our model undergoes only a rigid transformation, the deformation of each point on the shape is computed as a linear blend of multiple transformations through linear blend skinning. As a result, the local surface deformation is not necessarily rigid, enabling the representation of certain non-rigid deformations such as muscle stretching, as demonstrated in the object model shown in our **Supplementary Video**.
>
> Secondly, our skeleton representation is sparse, which is similar to SMPL [Loper et al. 2025] for the human body and SMAL [Zuffi et al. 2017] for quadrupeds. Due to the sparsity of the skeleton, the deformation model cannot directly model high-frequency deformation details, such as clothing wrinkles or hair dynamics. If the 2D motion prior contains such details, our method will capture the major pose change while ignoring these fine details.
>
> Finally, to model such high-frequency details, one approach is to design a denser skeleton, which is commonly used in character design for games or anime. For example, in NeuroSkinning [Liu et al. 2019], they designed skeletons for the character’s hair, dress, ribbons and so on. These additional bones provide more degrees of freedom to approximate higher-frequency deformation details. Another approach utilizes physical simulation to model the deformation of such soft materials. For example, Digital Salon [He et al. 2024] adopted the physical tool to model realistic hair movement. These represent
> another open and active area of research.
>
>   [He et al. 2024]: He et al. Digital Salon: An AI and
> Physics-Driven Tool for 3D Hair Grooming and Simulation. Siggraph Asia, 2024.

---

> > ### Comment · Reviewer_YFkN · 2025-11-25
> >
> > Thanks for the clarifications. I would like to maintain my score of 6.
> >
> > Although the authors have proposed ways to introduce more freedom for secondary motions, such methods are not part of this paper. The approach proposed in this work is still limited in modeling the dynamics in hair and ears since the bones are missing.

---

> > > ### Author Response · Authors · 2025-11-26
> > > **Response to Reviewer YFkN**
> > >
> > > Dear Reviewer **YFkN**,
> > >
> > > Thanks for your detailed reply and the positive rating on our work. However, we **kindly disagree** with the views of our method's limitations. First, we emphasize that our approach focuses on modeling articulated objects using a **skeleton-based deformation field**, while modeling high-frequency deformation details, such as hair, represents **a distinct research topic**, as noted in [He et al. 2024]. Additionally, limitations such as unrigged ear details in some models stem from limitations inherent to existing skeleton extraction methods, rather than from our proposed framework. Improved skeleton extraction techniques would naturally resolve this constraint.
> > >
> > >
> > > Best regards,
> > >
> > > The Authors

---

### Official Review · Reviewer_Esm6 · 2025-10-27

**Soundness:** 2
**Presentation:** 3
**Contribution:** 2
**Rating:** 4
**Confidence:** 4

**Summary:**

This paper tackles the task of rigging and animating template-free articulated objects represented by 3D Gaussians. The paper adopts a Mesh-Gaussian hybrid representation for articulated objects, enhancing the visual results of animated results using the continuity of mesh. It then learn motion-coherent skinning weights by leveraging motion priors from visual foundation models trained on large-scale 2D video datas. The proposed method outperform baselines.

**Strengths:**

1. The paper achieves better results than baselines by introducing additional drag-based image editing priors and optimization process.
2. The paper is overall well-written and easy to follow.
3. The idea of using drag-based image prior sounds reasonable to me since it can avoid the reliance on direct 3D supervision.
4. I appreciate the demonstration video.

**Weaknesses:**

1. The drag prompt of the image editing is generated by repose the mesh, whose skeleton is actually extracted by other feed-forward prediction method. Therefore, the method assumes that the base skeleton extracted is 'almost' right. Otherwise, the drag prompt cannot be generated reasonably.
2. The enhancement compared with baseline does not seem much as shown in Tab. 1.
3. The main focus of this paper is to use 3D Gaussians as the representation for rendering for higher visual quality. But with the development of Image-to-3D models like Hunyuan and TRELLIS, I just wonder what will happen if the rigging is operated directly upon the mesh. Some experiments regarding this would be helpful, also directly using mesh is more suitable for downstream usages in animation or game industry.
4. All examples shown in the paper are from synthetic datasets, some real-world examples would be beneficial.

I will be glad to raise my rating if my concerns are well addressed.

**Questions:**

See above.

---

> ### Author Response · Authors · 2025-11-22
> **Response to Reviewer Esm6**
>
> ```
> 1. The drag prompt of the image editing is generated by repose the mesh, whose skeleton is actually extracted by other feed-forward prediction method. Therefore, the method assumes that the base skeleton extracted is 'almost' right. Otherwise, the drag prompt cannot be generated reasonably.
> ```
> **Response**:
> Yes, we rely on skeletons extracted using existing methods. However, we have a certain tolerance for skeleton quality. We have added the dependency analysis of the skeleton in Appendix A.3 of the updated PDF file. When the skeleton contains slightly inaccurate offsets (Fig. 10 (1)) or redundant bones (Fig. 10 (2)), we can still obtain reasonable skinning weights. Our method can also predict reasonable skinning weights for partially accurate skeletons, see (3) and (4) in Fig. 10. However, when the skeleton is completely wrong (Fig. 10 (5)), we cannot handle it. Moreover, the sparse structure of the skeleton makes it easier to acquire than skinning weights. For example, transferring skeletons across similar structures or performing manual fine-tuning is efficient and feasible. This makes it easier to rapidly improve the performance of the skeleton prediction model. However, obtaining ground-truth dense skinning weights is highly challenging, as it often requires professional designers to expend considerable time and effort. Therefore, our method also aims to provide a practical automated skinning solution by leveraging rich 2D data priors to learn skinning weights for 3D data, thereby supporting the construction of large-scale datasets for automatic rigging tasks.
>
>
> ```
> 2. The enhancement compared with baseline does not seem much as shown in Tab. 1.
> ```
> **Response**:
> Since our task is to rig a skeleton to a static object and estimate skinning weights, rather than perform dynamic reconstruction from a given video. We can only adjust the skeleton pose to best match the target motion sequence. However, due to the lack of perfect pixel-level alignment between the rendered video and the ground-truth video, the rendering accuracy of all methods is somewhat limited. To facilitate a clearer comparison between the methods, we provide more visualization results in the updated Supplementary Video and in the Appendix (Figures 13 and 14 in the updated PDF file). As shown, the visual quality achieved by our method is significantly superior to that of the compared approaches.
>
> ```
> 3. The main focus of this paper is to use 3D Gaussians as the representation for rendering for higher visual quality. But with the development of Image-to-3D models like Hunyuan and TRELLIS, I just wonder what will happen if the rigging is operated directly upon the mesh. Some experiments regarding this would be helpful, also directly using mesh is more suitable for downstream usages in animation or game industry.
> ```
> **Response**:
> In our framework, it is entirely feasible to replace the 3D Gaussian representation with a textured mesh and use differentiable rendering to learn the skinning weights. However, our primary objective is to rig 3D Gaussians for high-quality and efficient rendering and animation. With the introduction and development of 3D Gaussian splatting, 3D Gaussians have demonstrated strong capabilities in efficient and high-quality rendering. In the future, more tasks may adopt this representation. However, existing rigged 3D datasets and rig prediction methods, such as UniRig, Magicarticulate, and Puppeteer, are predominantly dependent on mesh-based representation. In contrast, there is a scarcity of rigging methods for static, template-free object represented by 3D Gaussians. Therefore,
> our work aims to fill this gap and provide a technical method of constructing data to support the development of large-scale models for 3D Gaussian rigging. Additionally, we evaluated the animation results on 3D data generated by TRELLIS in Sec. 5 of the updated PDF file. As shown in Fig. 5 and the updated **Supplementary Video**, our method also works well with data produced by generative models.
>
>
> ```
> 4. All examples shown in the paper are from synthetic datasets, some real-world examples would be beneficial.
> ```
> **Response**:
> Thank you for your suggestion. We have added experiments with real-world data captured by ourselves in Sec. 5 of the updated PDF file and present the visual comparisons in Fig. 5. In the **Supplementary Video**, we have also added new motion sequences for these examples, demonstrating the practicality of our method.

---

> > ### Comment · Reviewer_Esm6 · 2025-11-26
> >
> > Thanks for the detailed reply. I appreciate the experiments upon mesh and real-world scenarios. Thus, I am willing to raise my rating to 6.

---

> ### Author Response · Authors · 2025-11-26
> **Looking forward to your further assessment**
>
> Dear Reviewer **Esm6**
>
> Thank you for taking the time to review our manuscript and for your valuable feedback and recognition. We have carefully addressed all the comments and concerns raised, as reflected in our detailed responses and the revised manuscript and supplementary material.
>
> We are looking forward to your further assessment.
>
> Best regards,
>
> The Authors

---

### Official Review · Reviewer_9Rtg · 2025-10-31

**Soundness:** 3
**Presentation:** 3
**Contribution:** 2
**Rating:** 4
**Confidence:** 3

**Summary:**

This paper proposes A$^3$-GS, a new framework for animating articulated 3D objects grounded on a 3D Gaussian-Mesh hybrid representation. The pipeline starts with reconstructing the static representation and extract the skeleton through corresponding off-the-shelf methods. Main technical contribution lies in the following skinning weights learning stage, where the authors leverage the motion priors from drag-based image editing models as guidance. In particular, they automatically construct drag signals by moving the joints of skeleton to generate the dragged images, which serve as supervision signals to optimize skinning weights. A local rigidity regularization and refinement module further stabilize training and enhance surface smoothness. Experiments conducted on articulated object datasets exhibit superior deformation performance compared to state-of-the-art rigging methods.

**Strengths:**

* Effective Mesh-Gaussian hybrid representation: The Mesh–Gaussian hybrid structure elegantly merges the continuity of meshes with the rendering efficiency of 3D Gaussians, addressing the core limitation in Gaussian-based animation.
* Novel use of 2D priors: The adoption of drag-based image editing as a motion prior is both original and practical, enabling rigging without labeled 3D motion data.
* Comprehensive evaluation: Extensive comparisons against strong baselines (UniRig, Puppeteer, MIA, MagicArticulate) demonstrate consistent performance improvements across metrics.
* Robust regularization design: The use of ARAP (as-rigid-as-possible) loss and local refinement ensures smooth deformation and visually coherent animations.

**Weaknesses:**

* Underlying instability of 2D generative priors: The accuracy of skinning weight learning heavily depends on the behavior of the drag-based image editing model. When the editing model misinterprets motion semantics (e.g., head rotations), the generated supervision can become misleading.
* Limited motion diversity: Since only one joint is perturbed per training step, complex coordinated motions are not well represented during training, which could limit performance in highly articulated motions.
* Moderate technical novelty: The proposed pipeline largely integrates multiple existing methods (e.g., GOF for reconstruction, off-the-shelf skeleton extraction). The core contribution lies in the procedure for generating dragged multi-view supervision.
* Lack of efficiency analysis: As the framework is optimization-based, the authors should report the time required for the entire workflow to better assess its practical efficiency.

**Questions:**

* How robust is the drag-based supervision when extended to multi-joint motions or larger rotations beyond the trained range (±0.35π)?
* Have you experimented with fine-tuning LightningDrag on skeleton-conditioned data to improve its motion alignment?

---

> ### Author Response · Authors · 2025-11-22
> **Response to Reviewer 9Rtg**
>
> ```
> Underlying instability of 2D generative priors: The accuracy of skinning weight learning heavily depends on the behavior of the drag-based image editing model. When the editing model misinterprets motion semantics (e.g., head rotations), the generated supervision can become misleading.
> ```
> **Response**:
> The learning of skinning weight relies on a drag-based image editing model; but **any better drag-based image editing model can be embedded into our framework**. With the rapid development of visual fundamental models, better drag-based image editing models will significantly improve the performance of our method. In Sec. 6 (Conclusion and Discussion) of **the originally submitted version**, we have clearly explained that the editing model may slightly misinterpret the semantics of motion (e.g., head rotation) and have clearly pointed out that fine-tuning on 3D data would be a practical future work. Nevertheless, our method still achieved promising results, outperforming existing approaches
> and demonstrating the effectiveness of our framework. Our updated **Supplementary Video** also provides extensive dynamic video comparisons.
> ```
> Limited motion diversity: Since only one joint is perturbed per training step, complex coordinated motions are not well represented during training, which could limit performance in highly articulated motions.
> ```
> **Response**:
> We add perturbations to each joint individually to fully leverage the advantages of drag-based models, avoid the impact of erroneous generation results, and thus improve training stability. Furthermore, we **introduce local rigidity-aware refinement in Sec. 4.3 of the originally submitted version** to enhance the continuity of complex movements by perturbing multiple joints simultaneously. We also present more visualizations in Figs. 8,9 in the originally submitted version (Figs. 13,14 in the updated PDF file), and the updated **Supplementary Video**. From these rich examples, we can see that our method is robust to complex multi-joint movements.
> ```
> Moderate technical novelty: The proposed pipeline largely integrates multiple existing methods (e.g., GOF for reconstruction, off-the-shelf skeleton extraction). The core contribution lies in the procedure for generating dragged multi-view supervision.
> ```
> **Response**:
> Since our task is to directly obtain a 3D Gaussian representation with a rig of an articulated object from multi-view images, enabling animation and high-quality novel-view rendering. It is an extremely **challenging system-level** task. We propose a new framework to solve this complex problem, which necessarily incorporates multiple modules, including 3D reconstruction and skeleton extraction. In addition to the design of skinning weights, we also incorporated Mesh-Gaussian hybrid representation, which was used for **the first time** in skeleton-based deformation to reduce artifacts in Gaussian rendering during animation. Numerous visualization results also validated the effectiveness of our method.
> ```
> Lack of efficiency analysis: As the framework is optimization-based, the authors should report the time required for the entire workflow to better assess its practical efficiency.
> ```
> **Response**:
> We have added the running time details in Appendix A.1. The reconstruction phase takes approximately 40 minutes, which includes 30 minutes for mesh reconstruction using GOF and under 8 minutes for training the Mesh-Gaussian hybrid representation. The rigging phase requires about 50 minutes, including a few seconds for skeleton extraction, 8 minutes for generating the guided images, around 40 minutes for motion-pair-guided skinning learning, and 2 minutes for local rigidity-aware refinement.

---

> ### Author Response · Authors · 2025-11-22
> **Response to Reviewer 9Rtg**
>
> ```
> Q: How robust is the drag-based supervision when extended to multi-joint motions or larger rotations beyond the trained range (±0.35π)?
> ```
> **Response**:
> In Appendix A.3 of the updated PDF file, we added further analysis to the drag-based model, including simultaneously perturbing more joints (10%–50% of joints) and a wider range of rotation angles ($0.1\pi-0.7\pi$). We can see that larger rotation angles and more joint perturbations result in worse generated images. Our settings (single-joint perturbation, relatively small angles ($\leq 0.35\pi$)) produce more stable results while covering the common range of motion of the joints. From our numerical comparisons and extensive visualizations, the effectiveness of our method is also demonstrated.
>
> ```
> Q: Have you experimented with fine-tuning LightningDrag on skeleton-conditioned data to improve its motion alignment?
> ```
> **Response**:
> We did not fine-tune LightningDrag, and fine-tuning would improve the performance and would be a practical direction for the future, as we **explicitly stated** in Sec.6 (Conclusion and Discussion) of the **originally submitted version**. Moreover, our proposed framework can **flexibly utilize any better drag-based image editing model**. A better model will lead to better performance for our method, not limited by LightningDrag.

---

> ### Author Response · Authors · 2025-11-26
> **Looking forward to your further assessment**
>
> Dear Reviewer **9Rtg**
>
> Thank you for taking the time to review our manuscript and for your valuable feedback and recognition. We have carefully addressed all the comments and concerns raised, as reflected in our detailed responses and the revised manuscript and supplementary material.
>
> We are looking forward to your further assessment.
>
> Best regards,
>
> The Authors

---

> > ### Comment · Reviewer_9Rtg · 2025-11-26
> >
> > Thanks for the detailed clarifications.
> >
> > * Since the underlying instability of 2D generative priors can not be addressed easily, could the authors provide several failure cases plus in-depth analysis?
> >
> > * Mesh-Gaussian hybrid representation is not new in the field of 4D generation. DreamMesh4D (https://arxiv.org/abs/2410.06756) has explored deforming the Mesh-Gaussian hybrid representation based on sparse control nodes, which can be treated as simple skeleton.
> >
> > * Only runtime of proposed method is reported, would love to see the efficiency comparison with baseline methods.

---

> ### Author Response · Authors · 2025-11-27
> **Response to Reviewer 9Rtg**
>
> Dear Reviewer **9Rtg**,
>
> Thank you for your thorough reply. Below, we provide further explanations and clarifications.
>
> ```
> Since the underlying instability of 2D generative priors can not be addressed easily, could the authors provide several failure cases plus in-depth analysis?
> ```
> **Response**:
> We understand your concern regarding the instability of 2D generative priors. It is well-known that 2D generative priors suffer from poor controllability and stability, making it **a challenging task** to use them for supervising the **learning of fine-grained skinning weights**. It is precisely the great challenge of this task that makes our method particularly significant. To **improve controllability**, we have designed several strategies:
>  - We adopt a drag-based image editing model instead of a video generation model, designing projected skeletal joints as drag signals to enhance consistency between the skeleton and the shape. Furthermore, an editing mask is automatically constructed to ensure the stability of unedited areas.
>  - To address the limitations of the drag-based model, we designed a strategy involving localized perturbations of individual joints.
>  - To tackle potential robustness issues with small joints, we introduced a geometric loss to achieve better binding between fine joints and local surfaces.
>  - We adopt the ARAP loss to improve training stability and maintain the smoothness of the deformed surface.
>
> Thanks to these designs, we can **stably and automatically learn skinning weights** from potentially unstable and difficult-to-control generative priors. As a result, our method produces very few failure cases. The only limitation of motion prior we observed is some instability in head rotation, which mainly stems from the fact that the data used by the drag-based model tends to favor head turning over swaying. However, this issue occurs from frontal views and is partially alleviated from a side view, as shown in **Fig. 11** of the updated PDF file. Moreover, we can indirectly supervise the head points through correct guidance from neighbor joint perturbations, such as upper arm movement. Therefore, it does not introduce significant flaws or shortcomings. Furthermore, as previously mentioned, our framework can integrate with any future, more advanced drag-based image editing model, and fine-tuning with rigged data will improve the performance. Now, highly effective closed-source generative models already exist, such as Nano Banana and Adobe Firefly AI, whose impressive results demonstrate that 2D generative models are becoming more powerful, stable, and controllable. As the controllability and stability of these models improve, they will enable even better results within our existing strategy. This potential advancement could also relax current limitations, such as reducing constraints on joint rotation angles.
>
>
> ```
> Mesh-Gaussian hybrid representation is not new in the field of 4D generation. DreamMesh4D (https://arxiv.org/abs/2410.06756) has explored deforming the Mesh-Gaussian hybrid representation based on sparse control nodes, which can be treated as simple skeleton.
> ```
> **Response**:
> The task in DreamMesh4D is **completely different** from ours. DreamMesh4D is focused on
> 4D generation from 2D video, while our work models the static articulated objects with skeleton. Although the skeleton-based deformation field shares some similarities with control node-based deformation fields, our task scenario is **fundamentally different**. The control node-based deformation is not suitable as a simplified skeleton because a skeleton is semantically meaningful and can be directly used for animation, whereas control nodes lack semantic meaning and merely fit the motion from a given video. In contrast, our task allows for arbitrary animation after the rigging is completed.
>
>
> ```
> Only runtime of proposed method is reported, would love to see the efficiency comparison with baseline methods.
> ```
> **Response**:
>
> Since **no** method is exactly the same as our task, we can only compare the time for the skinning weight learning. Our method differs from the compared approaches in terms of the training paradigm: ours is a case-by-case learning approach, while the competitors are data-driven, learning-based methods. Our approach requires approximately 50 minutes for optimization, while the inference times of the compared methods are as follows: 1s for MIA, 4s for UniRig, and 40s for Puppeteer. Although these methods exhibit fast inference speeds, they demand substantial training time. For example, Puppeteer requires about 1 day and 6 hours of training on 8 NVIDIA A100 GPUs, whereas our method involves **no pre-training**. Moreover, our work primarily addresses the poor generalization of data-dependent methods caused by the scarcity of 3D data with rigging. Our approach can be utilized to **construct more datasets**, thereby advancing the development of data-driven methods.

---

> > ### Comment · Reviewer_9Rtg · 2025-11-28
> >
> > Thanks for the rebuttal. Most of my concerns have been addressed.

---

### Author Response · Authors · 2025-11-22
**General Response**

We thank all reviewers for their time and constructive comments. We have carefully reviewed and addressed the questions and suggestions raised by the reviewers and carefully revised the manuscript accordingly. Please refer to the updated PDF file, with revisions highlighted in red. Here, we summarize the key issues and concerns:

- **Task and contribution**

    We focus on reconstructing rigged models with 3D Gaussian representation from multi-view images of static template-free articulated objects (character type) to achieve high-quality rendering and animation. This **system-level task** is particularly challenging. To address this complex problem, we propose a novel framework that incorporates a Mesh-Gaussian hybrid representation and a motion-prior-based skinning weight learning approach. To the best of our knowledge, this is **the first work to utilize a Mesh-Gaussian representation to mitigate rendering artifacts of 3D Gaussians in skeleton-based animation**, and also **the first work to utilize vision-based priors for learning skinning weights** of template-free objects. Extensive qualitative results (refer to the Appendix and **Supplementary Video**) demonstrate the effectiveness of our method.


- **Limitation of drag-based image editing model**

    We design strategies such as single-joint rotation to fully leverage the advantages of LightningDrag, and have included additional performance tests in the updated PDF file. According to our experiments, our training strategy has achieved promising results, outperforming existing methods. Importantly, our framework **can flexibly integrate any improved drag-based image editing model**, and better models will further enhance our performance.

-  **Missing comparisons with Gaussian-based animation methods**

    Since existing Gaussian-based animation methods, such as RigGS, address a different task setting (monocular video of dynamic subjects for RigGS) and cannot be directly compared to ours, we have instead introduced several variants in Sec. 5 in the updated PDF file by replacing our Mesh-Gaussian hybrid representation with Gaussian representations like RigGS and ARAP-GS. Experimental results demonstrate that our method produces fewer rendering artifacts during the animation process.

-  **Sensitivity to skeleton extraction**

    We have added a discussion on the dependence of skeleton extraction in Appendix A.3 of the updated PDF file.

- **Lack of real-world examples**

    We have added the test on real-captured data in Sec. 5 of the updated PDF file.

- **Others**

    Our updated **Supplementary Video** showcases further visualizations, including new motion sequences from both real-captured data and generated data obtained by the generative model (TRELLIS), as well as more comparative results on synthetic data. These rich visualizations illustrate the effectiveness of our method.

We appreciate the reviewers' insights, which significantly helped us refine and strengthen our work.

---

> ### Author Response · Authors · 2025-11-25
> **Looking forward to your further comments**
>
> Dear **Reviewers**,
>
> Thank you for taking the time to review our manuscript and for your valuable feedback and recognition. We have carefully addressed all the comments and concerns raised, as reflected in our detailed responses and the revised manuscript and supplementary material.
>
> We sincerely appreciate your efforts and look forward to your further assessment.
>
> Best regards,
>
> The Authors

---

### Author Response · Authors · 2025-11-29
**Summary for the Area Chair**

Dear **Area Chair**,


We kindly summarize the key comments, responses, and follow-up discussions, hoping it will help the new Area Chair quickly capture the core concerns and responses.

We first thank all reviewers for their time and constructive comments.
Throughout the initial comments and subsequent discussions, the reviewers positively acknowledged the **superior performance** of our method than baselines (9Rtg \& Esm6 \& YFkN \& 1iVk), the **rationale and novelty** of introducing 2D motion prior for learning skinning weights (9Rtg \& Esm6 \& YFkN \& 1iVk), the **effectiveness brought by the Mesh-Gaussian hybrid representation** (9Rtg \& 1iVk), and robust regularization design (9Rtg).

The reviewers' concerns are as follows:
the scope of the application (YFkN) and methodological novelty (9Rtg \& 1iVk), our method's dependency on the drag-based image editing model (LightningDrag) and the instability of LightningDrag (9Rtg \& 1iVk), the sensitivity to skeleton extraction (Esm6 \& 1iVk), efficiency analysis (9Rtg), and so on. Moreover, reviewers suggested additional comparisons with Gaussian-based animation methods (1iVk) as well as experiments on real-world data (Esm6).
We have provided **a detailed elaboration** on the challenges of our task, the novelty of our methodology, and its contributions and scope of application. In addition, we have **conducted supplementary experiments and visualizations**, which are included in the **updated PDF file and Supplementary Video**, and have provided **detailed responses** to address all related concerns.
Ultimately, all reviewers replied that **their concerns had been resolved** and reviewer Esm6 is willing to **raise the rating** to 6 from 4.

In short, our method demonstrates clear advantages, and the concerns raised by the reviewers have been successfully addressed. We thank the reviewers for their insightful comments and the area chair for the effort and time.

Best regards,

The Authors

---

### Meta-Review · Area_Chair_AJga · 2026-01-06

**Summary:**

This paper proposes a method for automatically creating animatable 3DGS characters. The main novel insight that it shares is the idea of using large 2D drag-based motion priors to improve learning of skinning blend weights on mesh-based 3D GS characters along with regularization terms to maintain robustness to failure cases. The method shows small but consistent improvements in quality versus the initial geometry-based skinning weights by incorporating the drag-based method.

Four reviewers provided scores (4,6,6,6). While many of the reviewers' concerns were addresses in the rebuttal, many remained unaddressed. There were no strong champions of the work.

Considering all things carefully, the AC feels that while the idea of using drag-based image priors to aid in the learning of skinning weights is interesting and shows promise, the small incremental gains in accuracy shown numerically and visually (in the supplementary video), combined with the obvious robustness and generalization issues of the drag-based method for motion modeling and its significantly long run tines, lower the overall contribution of the work. Hence, the work is below the bar for acceptance at ICLR and hence, the AC recommends rejection.

**Reviewer Concerns:**

Concerns addressed:
* efficiency analysis provided
* tests in generated and real-world captures assets
* lack of comparisons to RigGS and ARAP-GS

Concerns not addressed:
* limited novelty and contribution
* sensitivity of the proposed method to drag-based method's accuracy and limited joint angle rotations
* sensitivity of the proposed method to the estimated bone rigs
* marginal improvements in accuracy
* limited generalization to characters only

**Reviewer Scores:**

1. Reviewer 9Rtg (Rating: 4: marginally below the acceptance threshold. But would not mind if paper is accepted)

The reviewer's main concerns were around the use of (a) 2D drag-based motion priors to inform the learning of skinning weights, which can be problematic when the drag-based method fails, (b) modeling of only simplistic motion (single bone rotations), (c) limited novelty: the method largely uses existing components for mesh-based Gaussian reconstruction and skeleton prediction and only introduces the drag-based skinning weights learning component and (d) lack of efficiency analysis: the time for optimization is not mentioned. The reviewer indicated that most of their concerns were addressed.

2. Reviewer Esm6 (Rating: 6: marginally above the acceptance threshold. But would not mind if paper is rejected)

This reviewer's main concerns were that (a) the drag points are computed using the estimated skeleton, which presumes that the skeleton is correct, (b) testing of the method on 3D assets generated using Hunyuan3D or TRELLIS and with real-world captured data. Most of the reviewers concerns were address and they raised their rating to 6.

3. Reviewer YFkN (Rating: 6: marginally above the acceptance threshold. But would not mind if paper is rejected)

This reviewer's main concerns were that (a) only results on animated 3D characters and no laptops, etc are addressed by the method and (b) only part-wise rigid motion can be modeled with the proposed approach and non-rigid motion such as hair and clothing is not modeled. Of these, only (b) was addressed by the author's response. The reviewer maintained their rating.

4. Reviewer 1iVk (Rating: 6: marginally above the acceptance threshold. But would not mind if paper is rejected)

This reviewer's main concerns were (a) limited novelty and contribution, same as what was raised by Reviewer 9Rtg and (b) lack of comparisons to RigGS and ARAP-GS. Of these, only (b) was addressed by the author's response. The reviewer maintained their rating.

---

### Decision · Program_Chairs · 2026-01-26

Reject